# Risk assessment in water resources planning under climate change at the Júcar River Basin

Sara Suárez-Almiñana[1], Abel Solera[1], Jaime Madrigal[1], Joaquín Andreu[1], Javier Paredes-Arquiola[1]

[1]Research Institute of Water and Environmental Engineering (IIAMA), Universitat Politècnica de València, 46022, Valencia, Spain

*Correspondence to*: Sara Suárez-Almiñana (sasual@upv.es)

**Abstract.** Climate change and its possible effects on water resources has become an increasingly near threat. Therefore, the study of these impacts in highly regulated systems and those suffering extreme events is essential to deal with them effectively.

This study responds to the need for an effective method to integrate climate change projections into water planning and management analysis in order to guide the decision-making taking into account drought risk assessments. Therefore, this document presents a general and adaptive methodology based on a modelling chain and correction processes, whose main outcomes are the impacts on future natural inflows, a drought risk indicator and the simulation of future water storage in the water resources system (WRS).

This method was applied in the Júcar River Basin (JRB) due to its complexity and the multiannual drought events it suffers recurrently. The results showed a worrying decrease of future inflows, as well as a high probability ($\approx 80\%$) of being under 50% of total capacity of the WRS in the near future. However, the uncertainty of the results was considerable from mid-century onwards, indicating that the skill of climate projections needs to be improved in order to obtain more reliable results. Consequently, this paper also highlights the difficulties of developing this type of methods, taking partial decisions to adapt them as far as possible to the basin in an attempt to obtain clearer conclusions on climate change impact assessments.

Despite the high uncertainty, the results of the JRB call for action and the tool developed can be considered as a feasible and robust method to facilitate and support decision-making in complex basins for future water planning and management.

## 1. Introduction

The studies related to the possible effects of climate change on social, environmental, and economic frameworks have increased exponentially in recent decades. The main reason for this increase is the need to improve the adaptability of society and the capacity to manage risk, which was recognized by governments, scientists, and decision-makers at the World Climate Conference in 2009 and led to the creation of the Global Framework for Climate Services (GFCS) (Hewitt et al., 2013).

In fact, climate services have evolved over time to reach the wide variety of data that is available today, at the global (e.g.,

CORDEX - Coordinated Regional Climate Downscaling Experiment, https://www.cordex.org/), continental (e.g., SWICCA - Service for Water Indicators in Climate Change Adaptation, http://swicca.eu/) or national level (e.g., AEMET - State Meteorological Agency in Spain, http://www.aemet.es).

Normally, seasonal forecasts and climate projections are freely accessible through Internet portals, but the massive amount of data provided needs an advanced knowledge for their extraction. Therefore, some portals at continental and national level

facilitate the process of selecting models and variables by filtering them according to the fitting to the area and the user's needs (meteorological and hydrological variables, indicators, graphs, tables, etc.).

According to van den Hurk et al. (2016), climate services are essential to boost innovation in the water sector and increase its capacity to adapt to climate change. Hence, this big offer presents the opportunity to develop new tools or to improve the current ones incorporating climate projections in water management to extract useful information adapted to specific sectoral

needs (Hewitt et al., 2013). That is exactly what we aim to do in this study, proposing a general methodology inspired on the work of Suárez-Almiñana et al. (2017) to integrate climate projections in the decision process throughout a model chain for water management and drought risk assessments, where the future impacts on inflows and water resources are evaluated.

However, developing new methods is not easy, especially if it is for a long-term range, since anticipating responses to extreme events in a solid decision-making context for a distant future is challenging (van den Hurk et al., 2016). In addition,

van den Hurk et al., (2018) ensure that there is a gap between the spatial and temporal scales of the models versus the scales needed in applications and also highlight the need of tailoring climate results to real-world applications. These issues, among many others, may be the reason why so little climate action is taking place despite the wider knowledge of climate change (Naustdalslid, 2011).

Therefore, it seems that some issues need to be resolved in order to move forward in the process of developing these new

methods. The selection of projections and how to handle them correctly are part of these issues, since the inherent uncertainty of projections normally determines its use in practice (Lemos and Rood, 2010). In this sense, some authors recommend working with the ensemble (Stagl and Hattermann, 2015), since increasing the number of ensemble members reduce the sampling uncertainty (Collados-Lara et al., 2018; Thompson et al., 2017). Another option is differentiating between the Representative Concentration Pathways (RCPs) implied in the study (Barranco et al., 2018; Marcos-Garcia et

al., 2017) to consider the impacts related to the emission scenarios. However, working with only one ensemble member is not advisable, since the results can lead to erroneous conclusions due to the extreme values (Collados-Lara et al., 2018).

The need to reduce the uncertainty or increase the skill of these data is also a recurrent topic, but the dispersion of the ensemble members (EMs) is a fact over the world (Stagl and Hattermann, 2016; Chatterjee et al., 2018; Suárez-Almiñana et al., 2020), which would hamper the impact simulations (Teutschbein and Seibert, 2013) and influence the reliability of final

results, making decision-makers reluctant to consider these data for water management. The application of correction processes might be a solution to this problem, but these corrections may not provide a satisfactory physical justification (Ehret et al., 2012; Suárez-Almiñana et al., 2017) and it makes more difficult their inclusion in real-world applications.

Here is where the main improvement of the proposed methodology is focussed, the characterisation of future inflows, where correction and adjustment processes are applied to the ensemble in order to strictly adapt it to the case study in an attempt to reduce the uncertainty of simulated flows. Consequently, this step is also related to the proper calibration of the models involved in the modelling chain, which makes easier the complementation of management and risk assessments. All these efforts are related to the aim of obtaining more reliable results for decision-makers to trust these types of tools and to integrate them in the River Basin Management Plans (RBMP).

In fact, our study was focused in the east of Spain, the Júcar River Basin (JRB), where the inclusion of climate change assessment in the RBMP is mandatory, but it is not considered in the decision-making yet.

Thus, the need for an effective methodology that integrates the climate change projections to guide the decision-making is notable in this country and probably in many others. For this reason, the main objective of this study is to provide an answer for some of the above-mentioned issues, where an adaptive tool is developed to support and help basin managers to cope with future extreme events such as droughts, which may be more frequent and intense in the future (CEDEX, 2017; Marcos-Garcia et al., 2017). In addition, testing this tool in the JRB may be challenging, since this basin is heavily regulated and has a high hydrological variability that leads to face recurrent droughts of several years. Hence, the scarcity problems are expected to increase and early decision-making guided by a more accurate impact assessment will be needed.

To this end, we rely on different modelling approaches that can be found in the next sections. First, the features of the case study are presented in Section 2. The general methodology is then described in Section 3 in a simplified manner, followed by its adaptation to the JRB, where the climatic and local data, the methods of adjustment and correction and the characteristics of the modelling chain are specified. The hydrological model is the first in this chain and it is part of the characterization of natural inflows. This model is followed by the management model (deterministic approach) and the stochastic and risk assessment models (probabilistic approach). After that, Section 4 introduces the results of the approaches mentioned above. First, the adjustment and correction of the data (meteorological or hydrological) and the outputs of these processes after the hydrological model are presented, allowing to estimate the impacts on future water resources. Next, the future water storage in the system and the drought risk indicator are presented as part of the deterministic and probabilistic approaches, respectively. Finally, the discussion section justifies all the partial decisions taken during the process and the conclusion section summarizes the main outcomes of this study.

## 2. Case study: The Júcar River Basin

The Júcar River Basin is located in the eastern part of the Iberian Peninsula (Fig. 1) and it is the main water resources system (WRS) of the Júcar River Basin District (JRBD). Its extension is around 22,187 km$^2$ and the average volume of water resources generated is around 1,605 hm$^3$/year (CHJ, 2015). The river is 512 km long and the main tributaries are the Cabriel, Albaida, and Magro rivers.

This is a semi-arid area due to the influence of the Mediterranean climate. The average precipitation is 475.2 mm/year, the average potential evapotranspiration (PET) is 926.6 mm/year and the annual average temperature is between 14 - 16.5 °C, reaching the maximum in summer (June, July, and August), the dry season. Moreover, the high hydrological variability of this basin leads to recurrent multiannual droughts with some periods of floods in between.

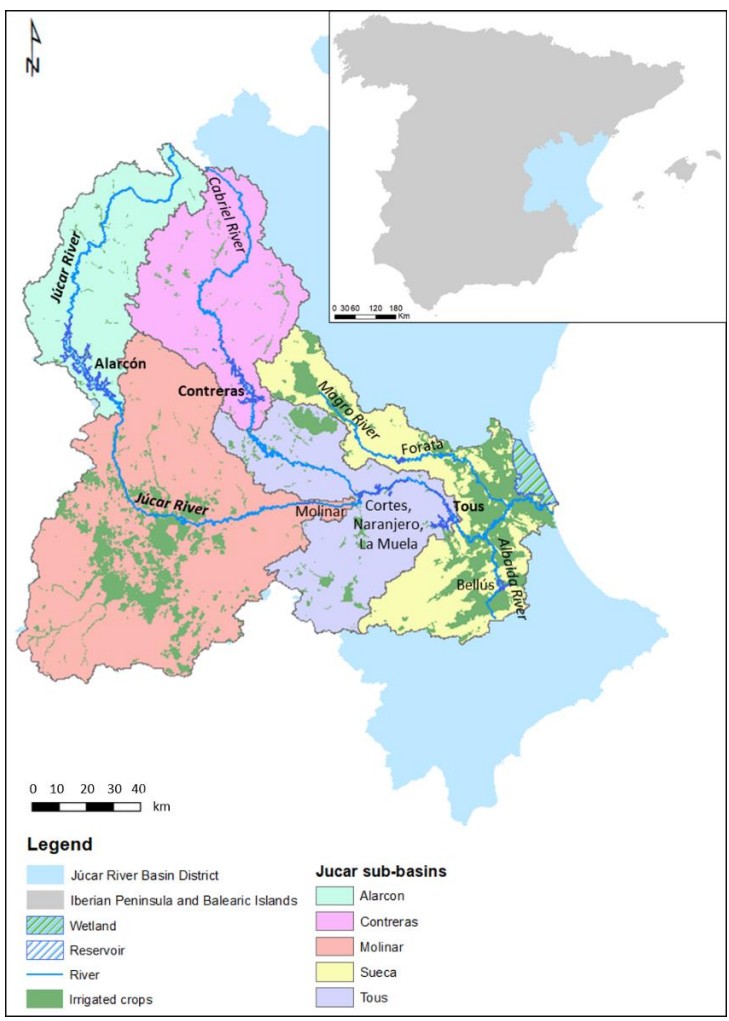

In addition to these hydrological features, consumptive demands are high (1,648.39 hm$^3$/year) (CHJ, 2015). The irrigated agriculture accounts for nearly 80% of water demand and other sectors (including urban supply) account for 20%.

The inland part of the basin is a mountainous area and the middle basin is a relatively flat area (high plain) that currently supports the major part of the irrigated agriculture ($\approx$ 100,000 ha). The lower basin lies in the coastal plain, which supports traditionally irrigated areas as well as more recent irrigated areas. There are permeable materials that allow rainfall infiltration to the aquifers of La Mancha Oriental (middle part of the basin, Molinar) and La Plana de Valencia (lower basin, Sueca), where groundwater is abstracted. In addition, there is an important wetland in the coastal area called l'Albufera de Valencia, which has an extension of 21,120 ha including a vast extension of rice crops.

Therefore, this combination of high water demand and hydrological variability forced to adapt by different management strategies, as water storage infrastructures, conjunctive use of surface and ground waters, and institutional and legal developments.

**Figure 1. Location of the Júcar River Basin District and the Júcar River Basin (divided in sub-basins) in Spain. Source: Confederación Hidrográfica del Júcar (CHJ, www.chj.es) and Instituto Geológico y Minero de España (IGME, http://www.igme.es/).**

Thus, this WRS is highly regulated, having several reservoirs, the more important are Alarcon (1,112 hm$^3$)

and Contreras (852 hm$^3$), which operate on a multi-year scale. On the other hand, Tous reservoir (314 hm$^3$) operates on an annual basis, storing the releases from upstream reservoirs and the inflows of the middle basin to supply the demands of this area. In addition, this reservoir is emptied in autumn to prevent floods from heavy rain events (Haro-Monteagudo et al., 2017). All the reservoirs are depicted in in Fig. 1, as well as how the JRB is divided in five sub-basins considering the reservoirs position and the hydrological features of the area.

Consequently, water stress is very high in this WRS, being the ratio between water demands and water resources around 90%. This means scarcity and leads to overexploitation of water resources, mainly during drought events, such as those reported in the periods 1981-1986, 1992-1995, 2005-2008, and 2013-2018. During these periods, some environmental and water quality problems arose, as well as high economic losses, but the conjunctive use of surface and ground waters proved to be a useful and robust tool against them. Nowadays, some other alternatives are used to avoid drought effects, such as drought emergency wells and wastewater reuse for agriculture (Haro-Monteagudo et al., 2017).

The institution in charge of the water management in the JRBD is the Júcar River Basin Authority (JRBA), which is also the responsible for the elaboration of the JRBDMP (Júcar River Basin District Management Plan) (CHJ, 2015) and the Drought Management Plan (DMP) (CHJ, 2018).

In this area, climate projections were not incorporated explicitly in the analysis made with the aid of Decision Support Systems (CHJ, 2015) for the last version of the JRBDMP, where climate change effects were assessed by reducing the natural inflows in a certain percentage (CEDEX, 2010) for the future hydrological cycles of management (6 to 18 years). More recently, climate projections were considered in the CEDEX (2017) report about the assessment of the climate change impact on water resources and droughts in Spain, where change rates of meteorological and hydrological variables were extracted for the main Spanish basins. The general conclusion for this district was the future decrease of water resources and the increase in the number of droughts and their intensity, but the results of this benchmark study have not yet been used in decision-making.

## 3. Material and methods

This section presents the general methodology, which is based on the integration of climate projections into a model chain for future management and drought risk assessments through the characterization of natural inflows followed by deterministic and probabilistic approaches. The model chain consists of hydrological, management, stochastic and risk assessment models.

In Fig. 2, this methodology is represented in a simplified manner. It was divided into three main parts that are closely related to each other, these are: i) the characterisation of natural inflows, where future inflows are extracted and some adjustments and corrections are applied to the ensemble to adapt it as much as possible to the current situation of the WRS; ii) the deterministic approach, where the future storage of the WRS is simulated and evaluated; and iii) the probabilistic approach, where the drought risk assessment is performed.

The main results that can be extracted from these sections are, respectively: impacts on future inflows, future water storage in the WRS and a drought risk indicator. All of them are complementary and may be very useful in the decision-making process.

The main improvement lies in the characterization of natural inflows, which is based on the extraction of inflows using the hydrological model and paying attention to some adjustment and/or correction processes. As can be observed in Fig. 2, the

input data for this model are precipitation and temperature time series from climate change projections, consisting of a reference period and a future period. In this sense, if the reference period is not fitted to the observed values of the local data, it may need a bias correction. To this end, we proposed two alternatives for this characterisation, called option A and option B. The main difference between these alternatives is the application of the bias correction before (option A) or after (option B) the use of the hydrological model.

In option A, the precipitation and temperature time series of the reference period are bias-corrected using the observed data. Then, this correction is extended to the future period series, which are introduced into the hydrological model to extract

the future inflows. Conversely, raw precipitation and temperature time series from climate projections are introduced into the hydrological model in option B. Afterwards, the hydrological outputs of the reference period are bias-corrected

using observed inflow data and this correction is extended to the future periods, thus obtaining the future inflows for this option.

These are simply two different ways of working with the same data in order to know which

alternative could be more reliable at the end of the process. Moreover, the good performance of the hydrological model in this step is essential, since it must to strictly represent the features of the basin.

Besides that, once the reference and future inflows

are extracted, they may be compared to extract the average change rates for the future, in other words, the effects of climate change on future inflows.

Afterwards, future inflows from A and B options

Figure 2. Methodology for the integration of climate change projections into the management and risk assessments to support decision-making.

(separately) are used in the deterministic approach, where they are introduced in the management model to simulate and

evaluate the future water storage of the WRS.

On the other hand, the statistical properties of future inflows (both options separately) are used in the probabilistic approach, in which the stochastic model generates multiple equiprobable series (taking into account these statistical properties) to perform the risk assessment. In this process, all the generated series are introduced in the risk assessment model, where the

management of the WRS is simulated for all of them and then, the management results are treated statistically to obtain a drought risk indicator related to the probability of reservoir storage in the WRS.

The steps of this methodology adapted to the JRB are detailed in the next sub-sections, where all the simulations were made taking into account the current conditions of the system, which may change in the future and affect water availability.

### 3.1 Climate change projections and historical local data

In this case, the climate projections from the SWICCA portal were selected for this study due to the good selection of Regional Climate Models (RCMs) for Europe it has available and the huge variety of data that can be downloaded at different temporal and spatial scales in a user-friendly format (.xlsx). This portal is a result of a Copernicus project that offers climate-impact data to speed up the workflow in the climate-change adaptation of water management across Europe.

Thus, precipitation and temperature time series of 9 RCMs from the RCPs 4.5 and 8.5 (IPCC, 2014) were downloaded at daily and catchment scales (mean area 215 km$^2$). These data came from the E-HYPE model (Hundecha et al., 2016), which uses global databases and Global Monitoring for the Environment and Security (GMES) satellite products as input data and then is forced by the European Centre for Medium-Range Weather Forecasts (ECMWF) and the Swedish Meteorological and Hydrological Institute (SMHI) to obtain meteorological, hydrological and another type of outputs for the entire continent (Hundecha et al., 2016; Suárez-Almiñana et al., 2017).

Table 1 shows the characteristics of the ensemble members used in this work. The reference period is 1971-2000 and the future periods are divided into 2011-2040 (near future), 2041-2070 (medium future), and 2071-2100 (far future). These data were obtained for the 5 sub-basins depicted in Fig. 1 and the last future period was reduced in 2 years due to the lack of data of two EMs.

**Table 1. Ensemble member characteristics from SWICCA portal. Modified from: http://swicca.climate.copernicus.eu/wp-content/uploads/2016/10/Metadata_Precipitation_catchment.pdf.**

| RCP | GCM | RCM | Period | Institute | Name of ensemble members |
|---|---|---|---|---|---|
| 4.5 | EC-EARTH | RCA4 | 1970-2100 | SMHI | SMHI_RCA4_EC-EARTH_rcp45 |
| | EC-EARTH | RACMO22E | 1951-2100 | KNMI | KNMI_RACMO22E_EC-EARTH_rcp45 |
| | HadGEM2-ES | RCA4 | 1970-2098 | SMHI | SMHI_RCA4_HadGEM2-ES_rcp45 |
| | MPI-ESM-LR | REMO2009 | 1951-2100 | CSC | CSC_REMO2009_MPI-ESM-LR_rcp45 |
| | CM5A | WRF33 | 1971-2100 | IPSL | IPSL-IPSL-CM5A-MR_rcp45 |
| 8.5 | EC-EARTH | RCA4 | 1970-2100 | SMHI | SMHI_RCA4_EC-EARTH_rcp85 |
| | EC-EARTH | RACMO22E | 1951-2100 | KNMI | KNMI_RACMO22E_EC-EARTH_rcp85 |
| | HadGEM2-ES | RCA4 | 1970-2098 | SMHI | SMHI_RCA4_HadGEM2-ES_rcp85 |
| | MPI-ESM-LR | REMO2009 | 1951-2100 | CSC | CSC_REMO2009_MPI-ESM-LR_rcp85 |

Then, the observed values of meteorological variables from the Spain02 v4 dataset (Herrera et al., 2016) were used as the historical local data. Spain02 is a gridded dataset of daily time series and 0.11° of spatial resolution that covers the Iberian Peninsula and the Balearic Islands for the period 1971-2010.

Currently, this database is used in this area due to its good performance (Pedro-Monzonís et al., 2016; Suárez-Almiñana et al., 2017; Madrigal et al., 2018; García-Romero et al., 2019) and it was needed for the bias correction of the climate projections (option A) and to test the calibration of the hydrological model. Thus, four points of each sub-basin (Fig. 1) were taken and averaged to obtain a representative time series per sub-basin (Madrigal et al., 2018) for the same reference period provided by the climate projections.

Another type of historical local data required in this analysis are inflow time series, which in this case are in natural regime (as if no anthropogenic modifications of the watercourse were applied) restored from observed data. These data were used in the calibration of the hydrological, management, and stochastic models, as well as for the bias correction in option B.

This dataset was provided by the JRBA for the period 1980-2012, which is used in the assessment of water resources reported in the JRBDMP, since the inclusion of previous years can lend to an overestimation of the available water resources
in the system after the "80s effect" (Pérez-Martín et al., 2013; Hernández Bedolla et al., 2019). This effect consists in a significant decrease of the average precipitations and inflows from 1980 onwards.

Henceforth we will refer to these data as natural or observed inflows.

### 3.1.1 Adjustment of the reference period

Within the climate projections was provided the reference period 1971-2000, but we proposed to reduce it to 1980-2000 so as to consider the "80s effect". As reported previously, the data series considered most suitable for working in the management of water resources of this basin are those observed from 1980 onwards, in this case from 1980 to 2012 (CHJ, 2015). Thus, the inflow series from the period 1980-2012, the reference period proposed (1980-2000), and the one provided by climate projections (1971-2000) were compared to determine their differences in terms of total water resources, as well as
to conclude if the proposed period is representing the current situation of the JRB. This process aims to avoid influencing the future with an excess of water resources through the application of the bias correction.

### 3.1.2 Bias correction

As the differences between climate projections and historical local data were notable in the reference period, a bias correction was advisable to adjust as much as possible the pan-European data to the regional scale. Hence, the correction of
245 precipitation and temperature variables was considered in option A and the inflows correction was considered in option B.

In this sense, one of the most reputed methods in literature is the quantile mapping, maybe because its application is relatively simple with good results, both for meteorological and hydrological variables (Grillakis et al., 2017; Manne et al., 2017; Teutschbein and Seibert, 2012). This method is based on the distribution function, which tries to keep the mean and

standard deviation of the reference series (Collados-Lara et al., 2018). In this case, it is a feasible approach since the observations are of similar spatial resolution as the EMs data (Maraun, 2013).

This process was applied using the R statistical software (https://www.r-project.org/) at daily (precipitation and temperature time series) and monthly timescales (inflows time series) by interpolating the empirical quantiles for variables of the reference period based on the package developed by Gudmundsson et al. (2012). First, the correction was made in the reference period using observed data and then it was extended to the future periods.

In addition, two quantitative statistics can be extracted to know the goodness degree of the RCMs concerning the observed data. Thus, the Nash Sutcliffe efficiency (NSE) (Nash and Sutcliffe, 1970) and Percent bias (PBIAS) (Gupta et al., 1999) values from corrected and non-corrected ensembles were obtained (Zambrano-Bigiarini, 2020) to know if the bias correction improved the fitting to historical data based on the performance ratings on daily time scale recommended by Kalin et al. (2010). The optimal values of NSE and PBIAS are 1 and 0 respectively and the proposed ratings are divided in: Very Good: $NSE \geq 0.7$, $|PBIAS| \leq 25\%$; Good: $0.5 \leq NSE < 0.7$, $25\% < |PBIAS| \leq 50\%$; Satisfactory: $0.3 \leq NSE < 0.5$, $50\% < |PBIAS| \leq 70\%$; Unsatisfactory: $NSE < 0.3$, $|PBIAS| > 70\%$.

### 3.2 Modelling chain

### 3.2.1 AQUATOOL Decision Support System Shell (DSSS)

To perform the modelling chain we employed the AQUATOOL DSSS (Andreu et al., 1996, 2009), which is a software widely used in the design of Spanish river basin plans, and also in many other basins abroad. It has several modules addressing different aspects of integrated water resources planning and management (WRPM) which are accessed from the same interface and are interconnected between them, an important feature to be considered in this study because the outputs of one model are the inputs of the others, as expected in a model chain.

The modules employed in this study were EVALHID (Paredes-Arquiola et al., 2012), SIMGES (Andreu et al., 2007), MASHWIN (Ochoa-Rivera, 2002, 2008) and SIMRISK (Sánchez-Quispe et al., 2001; Haro-Monteagudo, 2014; Haro-Monteagudo et al., 2017). These modules were used to build the hydrological, management, stochastic, and risk assessment models, respectively.

EVALHID module has available several rainfall-runoff models with different structural complexities and parametrizations, but all of them have been aggregated with semi-distributed applications at the sub-basin scale (García-Romero et al., 2019; Hernández Bedolla et al., 2019; Suárez-Almiñana et al., 2017).

SIMGES module is used to simulate the management of the WRS for water allocation. Here, a simplification of the WRS can be drawn using a friendly interface, where the databases related to all its elements (as reservoirs, contributions, demands, returns, aquifers, channels, environmental flows, etc.) can be filled along with the operating rules and the water use rights and priorities. All these features are considered to simulate the water allocation using an optimization algorithm for deficits minimization and maximum adaptation to the reservoir objective volume curves.

MASHWIN allows the building of multivariate stochastic models to generate multiple and equiprobable synthetic series, preserving the statistical properties of the original series for the generation. It is a complement for SIMRISK, since it needs a high number of flow series to perform the risk assessment.

SIMRISK uses the multiple generated series to extract probabilistic results on reservoirs storage and demand deficits among others. This tool can be used in the short, medium, and long term and its purpose is to inform the decision-makers about the probable state of WRS in the future. In this way, they can propose measures to minimize possible impacts and simulate different management scenarios to choose the most effective ones for reducing the impacts (Haro-Monteagudo, 2014).

### 3.2.2 Hydrological model

This model was employed to evaluate the amount of water resources produced in the basin using precipitation and PET time series from the ensemble as input data. The Hargreaves method (Hargreaves and Samani, 1985) was used to convert temperature into PET. In spite of the huge variety of methods with different skills to carry out this conversion (Milly and Dunne, 2017), its performance for this area is very valuable (Espadafor et al., 2011; Hernández Bedolla et al., 2019) and the data needed to apply it can be easily obtained.

In this case, the rainfall-runoff model HBV (Bergström, 1995) was selected to extract inflows from input data due to its good performance in this basin at daily scale after a proper calibration, which was performed by García-Romero et al. (2019) using two optimisation algorithms and the observed inflows from the period 1980-2007.

This model was run using bias-corrected time series of precipitation and PET in option A (Fig. 2), while in option B it was run using non-corrected data and then the output inflows were bias corrected before inserting them in the rest of the models of the chain.

Thus, corrected and non-corrected precipitation and PET were introduced in the HBV model to assess its performance in the reference period, and then generate future flows for the management and risk assessments. For both options, the simulation of future inflows was made using the time series from 2011 to 2098, in this way, initial conditions for all periods are conserved and maintained, as well as the tendency of future inflows.

In this case, the values of NSE and PBIAS statistics were also extracted to estimate the performance of the model run with Spain02 data to ensure its good calibration and then see if the bias correction improved the ensemble fitting to observed data. This time we based on the performance rating recommended by Moriasi et al. (2007) because we are comparing inflows at monthly time step. The ratings are divided in: Very Good: NSE ≥ 0.75, |PBIAS| ≤ 10%; Good: 0.65 ≤ NSE < 0.75, 10% < |PBIAS| ≤ 15%; Satisfactory: 0.5 ≤ NSE < 0.65, 15% < |PBIAS| ≤ 25%; Unsatisfactory: NSE < 0.5, |PBIAS| > 25%.

Afterwards, the future ensembles from each sub-basin, period and option were compared with their respective ensemble baselines (1980-2000) to evaluate the climate change impact on future flows. The average change rates of future periods were obtained from the ensemble mean, not counting the increment or reduction of previous periods.

### 3.2.3 Management model

On this occasion, a simplified model of the Júcar River WRS was used to simulate the future water allocation for this basin. The main elements of the WRS were integrated into this model, as well as the operational rules and all the features involved in the current management of the system (CHJ, 2015).

The most interesting result we can extract from this model for the current study is the future water storage for the whole system, which volume was considered as the sum of the Alarcon, Contreras, and Tous reservoirs (1796 hm$^3$). Thus, the entire period of future inflow series (2011-2098) from the previous step was used to run this model and extract those results for options A and B. In this way, the future evolution of storage values can be better observed to complement the results of the risk assessment.

### 3.2.4 Stochastic model

In this case a multivariate autoregressive model of first-order AR(1) was enough to generate the series after the time dependence parameter was calibrated using natural inflows from the 1980-2012 period. Then, this model was modified to adapt it for the generation of future series, since it was calibrated for the historical scenario. The statistical properties (mean and standard deviation) of future inflows obtained in the previous section (options A and B) were used for this purpose. Hence, based on these future statistical properties, the model generated 1,000 synthetic series per EM and future period (the three considered) to feed the risk assessment model. The more series we generate, the more statistically robust results at the end of the process (next step).

### 3.2.5 Risk assessment model

In this model, the water management of the system was simulated for all the series generated in the previous step, based on the Monte-Carlo method. Then, the management outputs were treated statistically to extract the drought risk indicator. This probabilistic indicator informs about the evolution of the water storage of the system for the ensemble and the three future periods. As in the previous case, the sum of volumes of the main reservoirs was considered as the total storage of the system.

### 4. Results

In this section, the ensemble mean and the range covered by all EMs are shown in the figures. We decided to work with the ensemble of both RCPs 4.5 and 8.5, since in this way the approximation to the most likely future scenario (the RCP 6.0) accorded in the Paris Climate Change Conference 2015 (Barranco et al., 2018) is possible. The RCP 6.0 is an intermediate scenario of those employed, but no projections were available for this scenario, so this is a way of approaching it and to simplify the process.

## 4.1  Analysis of variables and their bias correction

Regarding the proposal of adjusting the reference period, in Fig. 3 is depicted how the average annual inflows observed from the period 1980-2012 and the reference period we proposed (1980-2000) can be considered as equivalent (Suárez-Almiñana et al., 2020), while the reference period provided (1971-2000) has higher total inflows, which we want to avoid to have a good representation of the current situation of the JRB.

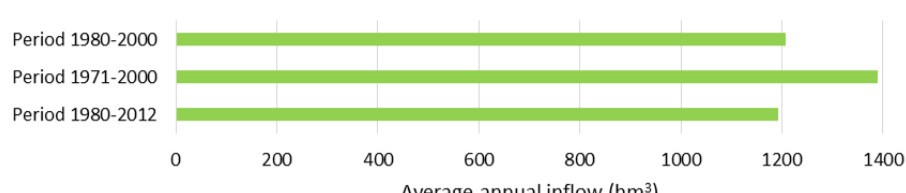

**Figure 3. Average annual inflows observed in the Júcar River Basin for different historical periods. Modified from Suárez-Almiñana et al. (2020).**

Thus, we proceed with the proposed reference period (1980-2000) to make the comparison between precipitation and temperature series of the ensemble and the observed data (Spain02). In this comparison, a general overestimation of temperature on the average year of this period and an underestimation of precipitation in most of the sub-basins was detected (Fig. 4). As these variables were not in the same line, the bias correction was applied to both variables.

While the overestimation of temperature disappeared after the application of this technique, the differences between the corrected ensemble of precipitations and the observed data were minimized (Fig. 4), as well as the average, but it is still overestimated in spring and summer. Moreover, Fig. 4 shows how the bias correction provided a little difference favouring some months and affecting others in Molinar and Tous sub-basins, but very subtly in both cases. However, all these differences can be assumed to obtain more reliable flows in the next step (Fig. 5). In addition, based on the performance rating proposed by Kalin et al. (2010), the values of the PBIAS statistic made Alarcon and Sueca sub-basin go from good to very good performances after the bias correction, while the other sub-basins did not change the very good status but the PBIAS values were more proximal to 0% (the optimal value). Despite this, the NSE values for all sub-basins of non-corrected series were unsatisfactory and the bias correction was not enough to go beyond this threshold value (0.3).

Then, this correction was extended to the future series and the corrected temperature time series were converted into PET (using the Hargreaves method) to prepare the data for the hydrological model.

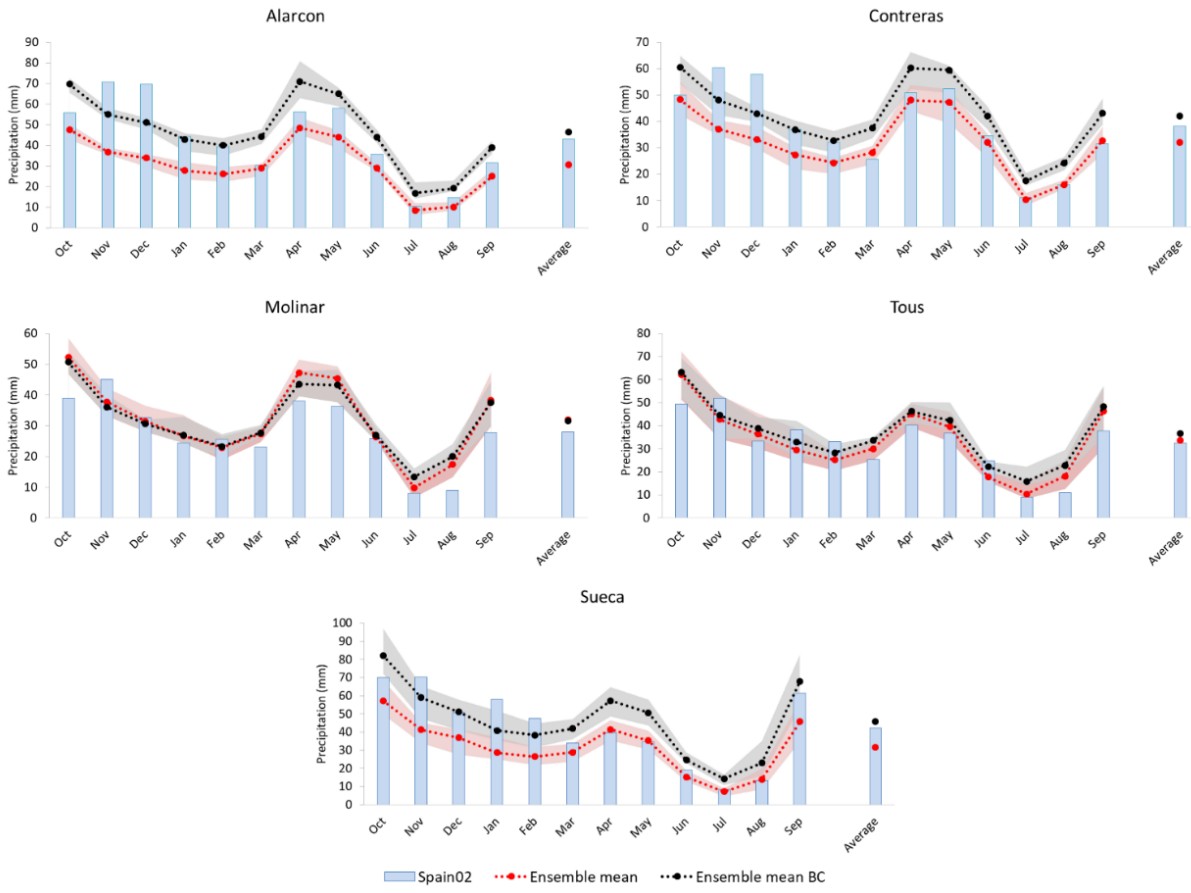

365

**Figure 4. Average monthly and yearly bias-corrected precipitation (Ensemble mean BC) compared to the non-corrected precipitation (Ensemble mean) and the historical data (Spain02 data) in the reference period 1980-2000, where the shaded areas represent the entire ensemble.**

## 4.2 Natural inflows characterisation

In this section, corrected and non-corrected precipitation and PET time series were introduced into the HBV model to assess its performance and then generate future inflows for the management and risk assessments. In the next sub-sections the results for option A and option B are presented.

### 4.2.1 Option A: HBV model simulation using bias-corrected data

First, the inflows obtained from the HBV model fed with meteorological historical data (HBV-JRB Spain02) were compared with the observed inflows to assess its performance and validate it for the JRB. This comparison is illustrated in Fig. 5,

where it can be seen how both data are generally close, as well as their averages, setting aside some differences that are likely due to its parametrization in the calibration process.

In order to assess the performance of the model, the NSE and PBIAS values were obtained for the case of the HBV-JRB Spain02 inflow series. Based on the performance ratings recommended by Moriasi et al. (2007), the NSE values showed very good and good performances for Alarcon and Contreras respectively (Table 2), while the values from the others sub-basins had an unsatisfactory performance. However, the same ratings but based on PBIAS values, shows how Contreras and Molinar have a very good performance, in Alarcon and Tous it performs good and it is satisfactory for Sueca.

Thus, we can say that the HBV model is more accurate in the headwaters basins (Alarcon and Contreras) where the main reservoirs are placed, a fact to be considered from the water management point of view. In this way, the apparent mismatch in the Sueca sub-basin is not relevant for the purposes of this study since it is located in the final stretch of the river, where there is no reservoir regulation available. In the case of Molinar and Tous, inflows were underestimated, but these differences were expected because these sub-basins are the most heavily regulated and difficult to simulate with hydrological models, mainly due to its close connection with the underground component. Despite these differences, the performance of the HBV model using historical data can be considered as acceptable and quite good due to the huge complexity of this basin. Thus, it was decided to continue with the study simulating the ensemble inflows for the reference and future periods.

In this case, Fig. 5 (middle part) was completed including the inflows from the corrected ensemble (HBV-JRB Ensemble mean A). There, it can be seen how HBV-JRB Ensemble mean A inflows are more or less in line with the observed inflows and its average, setting aside some differences that are likely due to the HBV mismatches and the precipitation overestimation during the spring months coming from the bias-corrected process. The rates of Table 2 show a worse performance than those obtained with the historical data, indicating that the fitting of the corrected ensemble to the historical period is not good enough despite the bias correction and the good calibration of the HBV model.

In the Alarcon sub-basin, the ensemble is underestimating river flows in January and February (as in Contreras), while it is overestimating them in spring months, which is likely related to the outputs of the bias correction process in these months. In the Molinar sub-basin, this ensemble has higher values than the HBV-JRB Spain02 inflows and they are closer to the observed ones. In the case of Tous inflows, they are overestimated and in the Sueca sub-basin, both inflow series overestimate observed river flows from November to January and the ensemble also overestimates spring flows, which may be due to the overestimation in corrected precipitation.

### 4.2.2 Option B: HBV model simulation using raw data and bias correction of flows

In this section, the raw precipitation and PET time series of the reference period were introduced into the HBV model to extract the non-corrected inflows (HBV-JRB Ensemble mean) and evaluate if the previous correction was worth it or not.

Looking at Fig. 5 (left) and Table 2, it is evident that a bias correction was needed on meteorological or hydrological data, since the non-corrected inflows are not representing the current situation of the basin, obtaining good performances only in Molinar and Tous sub-basins for PBIAS rates. These inflows of the reference period are highly underestimated in Alarcon

and Contreras and if this is extended to future flows, the conclusions on the impacts of climate change can be misleading and have a severe and false view of the future. Thus, in this part was decided to correct those inflows and see the differences between correcting data before and after running the hydrological model. These inflows were also corrected using the quantile mapping method and the improvement was notable, particularly in the average fitting (Fig. 5, right) and the ratings for the PBIAS values (Table 2). Despite this, there are some mismatches in accordance to the previous section (Fig. 5, middle and right), which are also captured by the NSE statistic. There are some underestimations in January and February in Alarcon and Contreras and spring months are also overestimated. However, in Tous and Molinar sub-basins the corrected inflows are more or less in line with the observed ones and in Sueca, December and May inflows are overestimated.

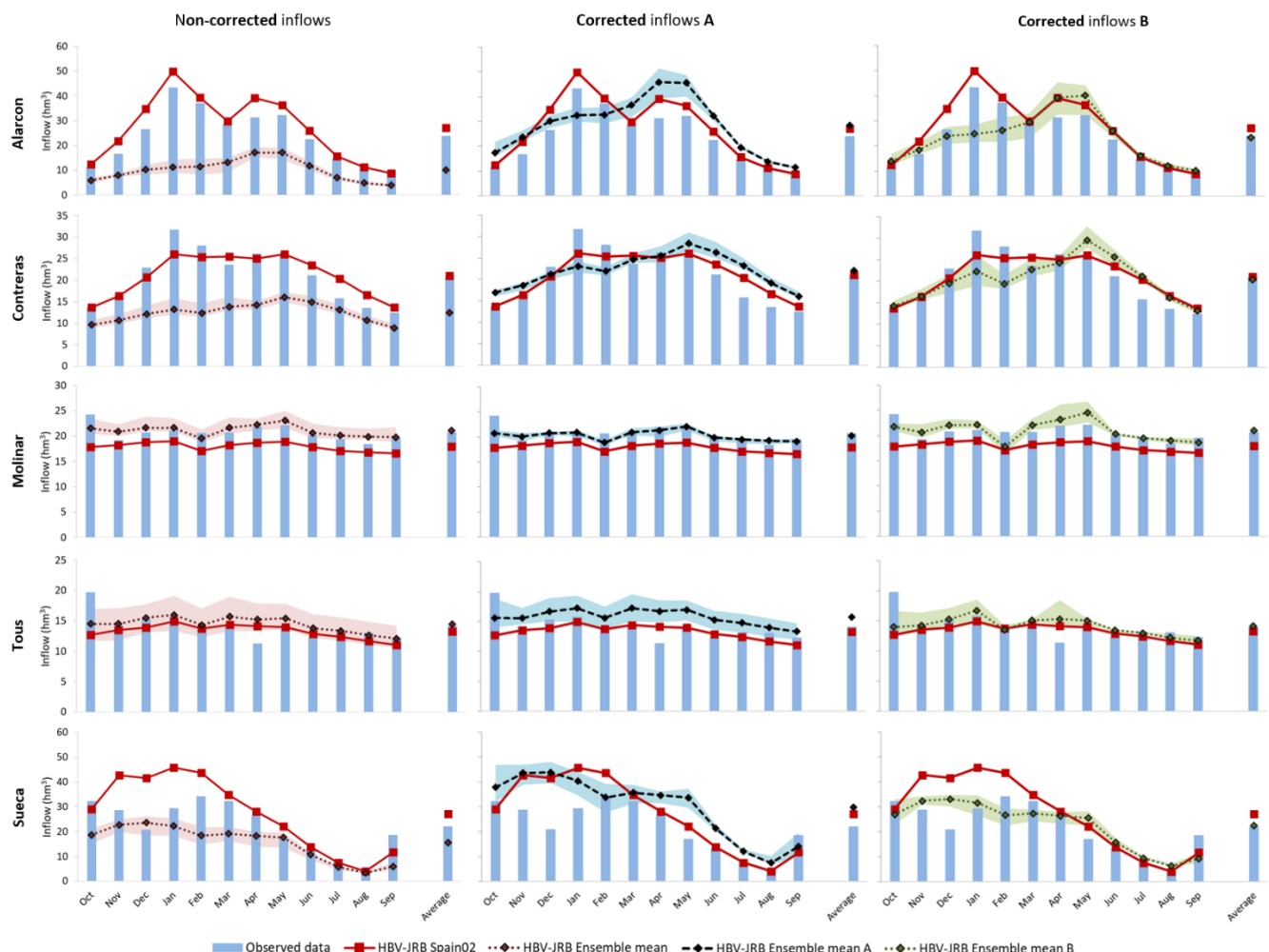

**Figure 5.** Average monthly and yearly inflows from the application of the HBV model using historical (HBV-JRB Spain02) and raw ensemble data (HBV-JRB Ensemble mean and shaded area) compared to the observed (Observed data) and corrected inflows (HBV-JRB Ensemble mean A, HBV-JRB Ensemble mean B and shaded areas) in the reference period 1980-200.

In general, these corrections can be considered as acceptable because non-corrected inflows are not an option to follow with
the process, mainly due to the underestimation of headwaters inflows. Moreover, at least the PBIAS ratings are better in the
corrected options. Thus, these corrections were extended to future inflows.

**Table 2. HBV-JRB model performance depending on simulated data and their PBIAS and NSE values based on the classification of the performance ratings recommended by Moriasi et al. (2007) for monthly time steps of streamflows. Where VG is a very good performance, G is good, S is satisfactory, and U is unsatisfactory.**

|  |  | Alarcon | Contreras | Molinar | Tous | Sueca |
|---|---|---|---|---|---|---|
| **HBV-JRB Spain02** | PBIAS (%) | G | VG | VG | G | S |
|  | NSE | VG | G | U | U | U |
| **HBV-JRB Ensemble mean** | PBIAS (%) | U | U | VG | VG | U |
|  | NSE | U | U | U | U | U |
| **HBV-JRB Ensemble mean A** | PBIAS (%) | S | VG | VG | G | U |
|  | NSE | U | U | U | U | U |
| **HBV-JRB Ensemble mean B** | PBIAS (%) | VG | VG | VG | VG | VG |
|  | NSE | U | U | U | U | U |

### 4.2.3   Impact on future inflows

In Fig. 6, the impacts on future inflows are depicted per sub-basin, period, and option, as well as for the whole JRB.

As expected from other studies, the average year inflows decrease over future periods, but the average change rates differ
from sub-basins and approach. If we compare both results (Fig. 6, top and middle), the reductions in the headwaters are
important but more drastic in Alarcon for option A, where these change rates reach in average -20% for the far future (Fig. 6,
top right). However, the drastic decrease was found in the Molinar sub-basin of option B, which reaches -21% as average in
the far future (Fig. 6, middle right). Then, the inflows behaviour in Tous is remarkable (in both cases), since there is a large
inflow increase in the near and medium futures (mostly in option B) that later decreases in the last period. The reason for this
increase may be the high influence this sub-basin has from the underground component. Moreover, increasing contributions
to this sub-basin have been observed in recent years (Hernández Bedolla et al., 2019), which may continue and be translated
into more contributions to this sub-basin until the second period.

However, the Sueca sub-basin has very similar decreases in both options, reaching -18% as average in the last future period.
The same happens if we look at the JRB as a whole (Fig. 6, bottom), the differences between using A and B approaches are
minimal, reaching about 3% as average in the near future, -3% in the middle future and -12% in the far future.

Hence, we can say that there are important decreases in the headwaters, which may be a great challenge for future
management because in these areas is where the main reservoirs are located. Moreover, the sharp reductions in Molinar and

Sueca sub-basins are also concerning. In Molinar, reduced inputs may lead to a decrease in infiltration into the main aquifer in the basin (La Mancha Oriental), while in Sueca this may increase the demand and pressure on irrigation campaigns, since this is the area where the most of the irrigated crops are located (Fig. 1).

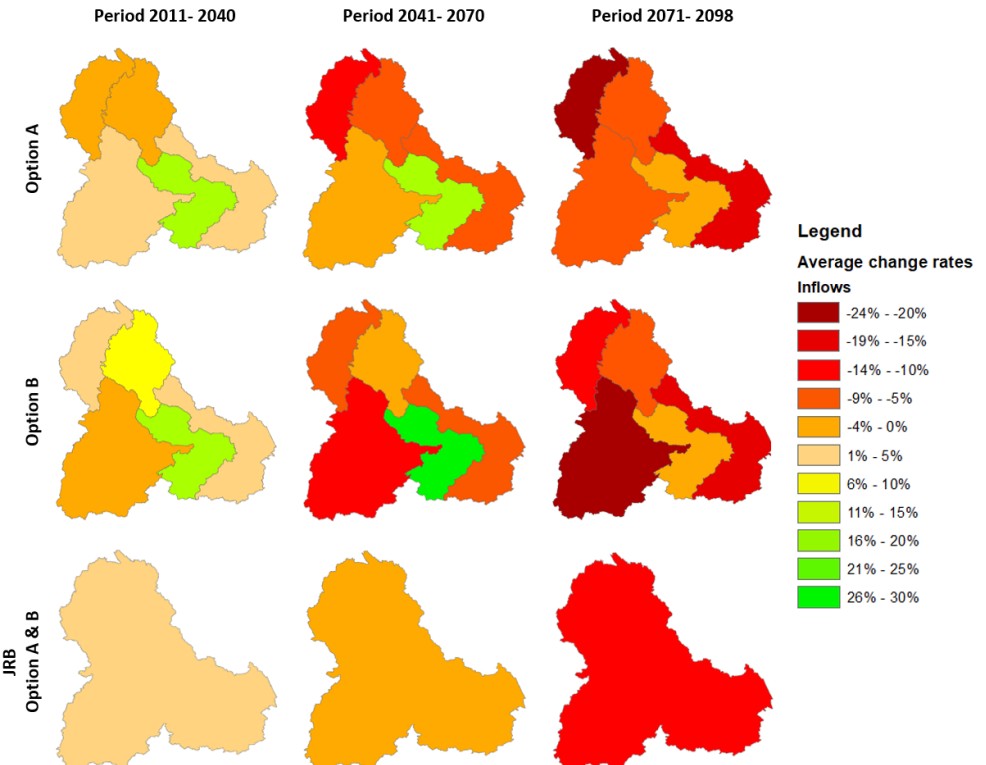

**Figure 6. Average change rates of inflows per sub-basin and the whole Júcar River Basin (bottom) for the future periods 2011-2040, 2041-2070, and 2071-2098, distinguishing between options A (top) and B (middle).**

### 4.3 Future water storage in the system

In Fig. 7, the future storage volumes for the ensemble of both options, A and B, were represented taking into account the total capacity of the system (1796 hm$^3$). These results were simulated with the water allocation model using future inflows from the previous section.

In general, the mean values from option B (Fig. 7, bottom) are lower than those from option A (Fig. 7, up), which may result in worse climate change impacts from the middle century onwards. In addition, the frequency area of the EMs (lighter shaded area) shows the same conclusion, while in option A most EMs coincide in the upper parts of the storage volume with a couple of critical periods, option B describes a more critical situation with several and recurrent drought periods from the second period onwards. However, the ensemble results (darker shaded areas) occupies practically the whole field of stored volume in the basin, indicating a huge uncertainty for the future.

The dispersion of option A is less intense (see shaded area), mainly due to the minimum values of the EMs, which are higher than those of option B, especially until the mid-century. Therefore, the future conditions presented in option A provide more optimistic results, but their large dispersion makes results not reliable for the future, as in the case of option B.

Thus, these deterministic results have to be completed and complemented with probabilistic outcomes from the risk assessment in order to be more trustable from the point of view of decision-makers.

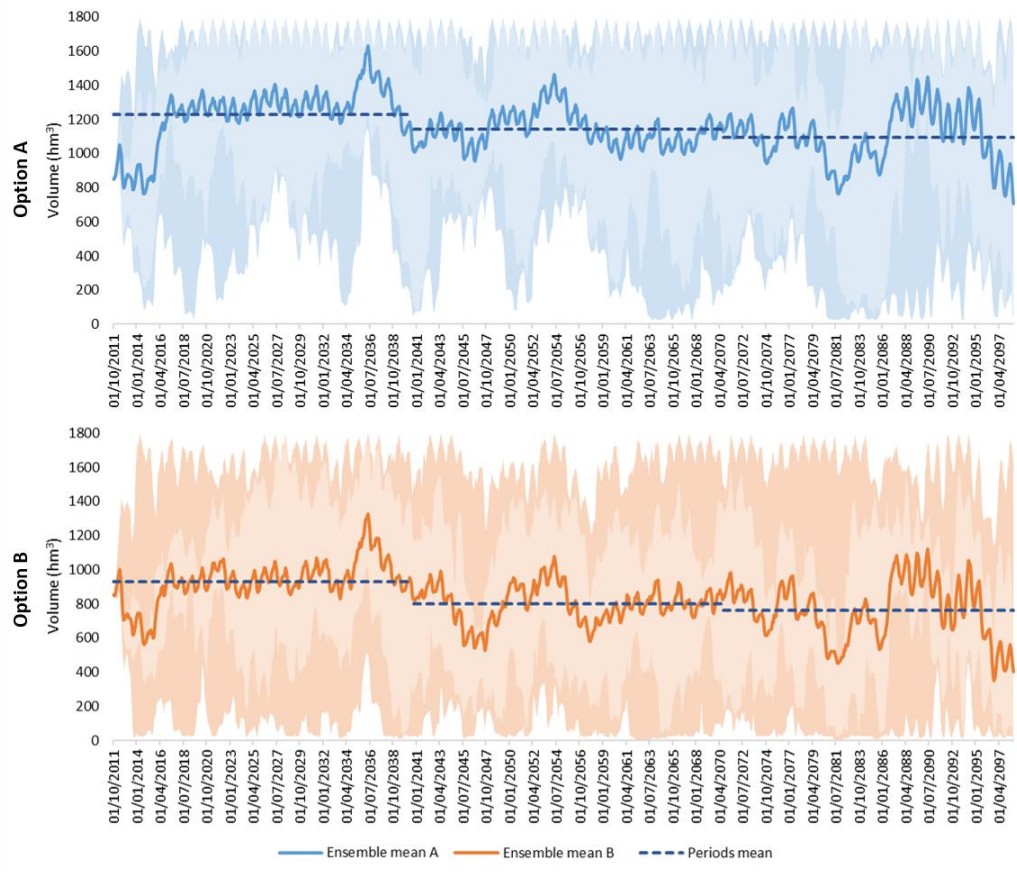


**Figure 7. Evolution of the water storage in the Júcar WRS for the ensemble of options A (up) and B (bottom) in the future period 2011-2098.**

## 4.4   Drought risk indicator

After the generation of multiple synthetic inflow series in the stochastic model and their integration in the risk assessment model, the probabilistic evolution of the reservoir storage in the system was extracted in form of risk indicator, which can be seen in Fig. 8 for both options A and B. There, the ensemble mean indicator for each future period and approach is represented, where the total capacity of the system (1,796 hm$^3$) was divided into 10 equal intervals and the probability of being in each interval was displayed for each period.

The probabilities are very similar in all future periods of both alternatives. In both options, the probabilities of being under the 50% of total capacity (898 hm$^3$, medium green colour) is about 80% in the near future, but these probabilities are around 70% and 60% in the medium and far future respectively, a little higher for option B. This may lead to the conclusion that the probabilities of being at lower intervals are decreasing over the periods despite the average inflow reductions obtained in Fig. 6 and the mean future volumes observed in Fig. 7, but this is due to the greater probability of falling in any interval

(≈10%) as time goes on. This indicates a high uncertainty for the future, since there is a large variation in future simulated storage volumes, as was expected from the shaded areas depicted in Fig. 7.

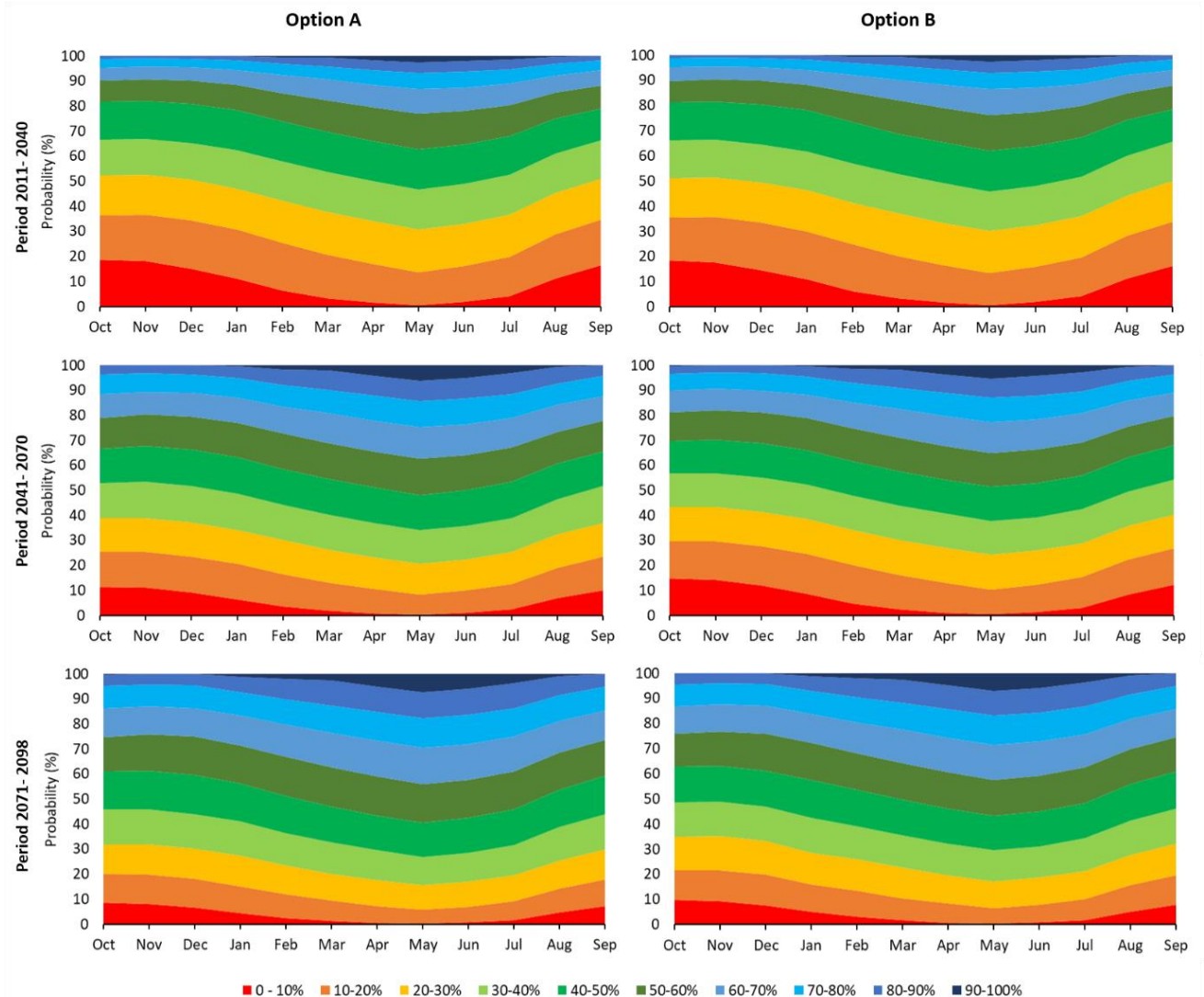

**Figure 8. Drought risk indicator of the ensemble mean per option (A and B) and future period (2011-2040, 2041-2070, and 2071-**
**2098).**

Looking at the indicator results, we decided to pay attention to the exceedance probabilities of March and September (Fig. 9) as these months coincide with the start and the end of the irrigation season, respectively. In addition, those results for September also inform about the final state of the system for each future period, coinciding with the end of the hydrological year.

In the first period, the range of exceedance probabilities covered by the ensemble is very tight in both months, coinciding more or less with the ensemble mean of both approaches, while in the other periods this range is wider due to a higher dispersion of the EMs. In general, ensemble results from option A show higher probabilities of exceeding higher storage volumes in both months, as was expected from results shown by Fig. 7 and Fig. 8.

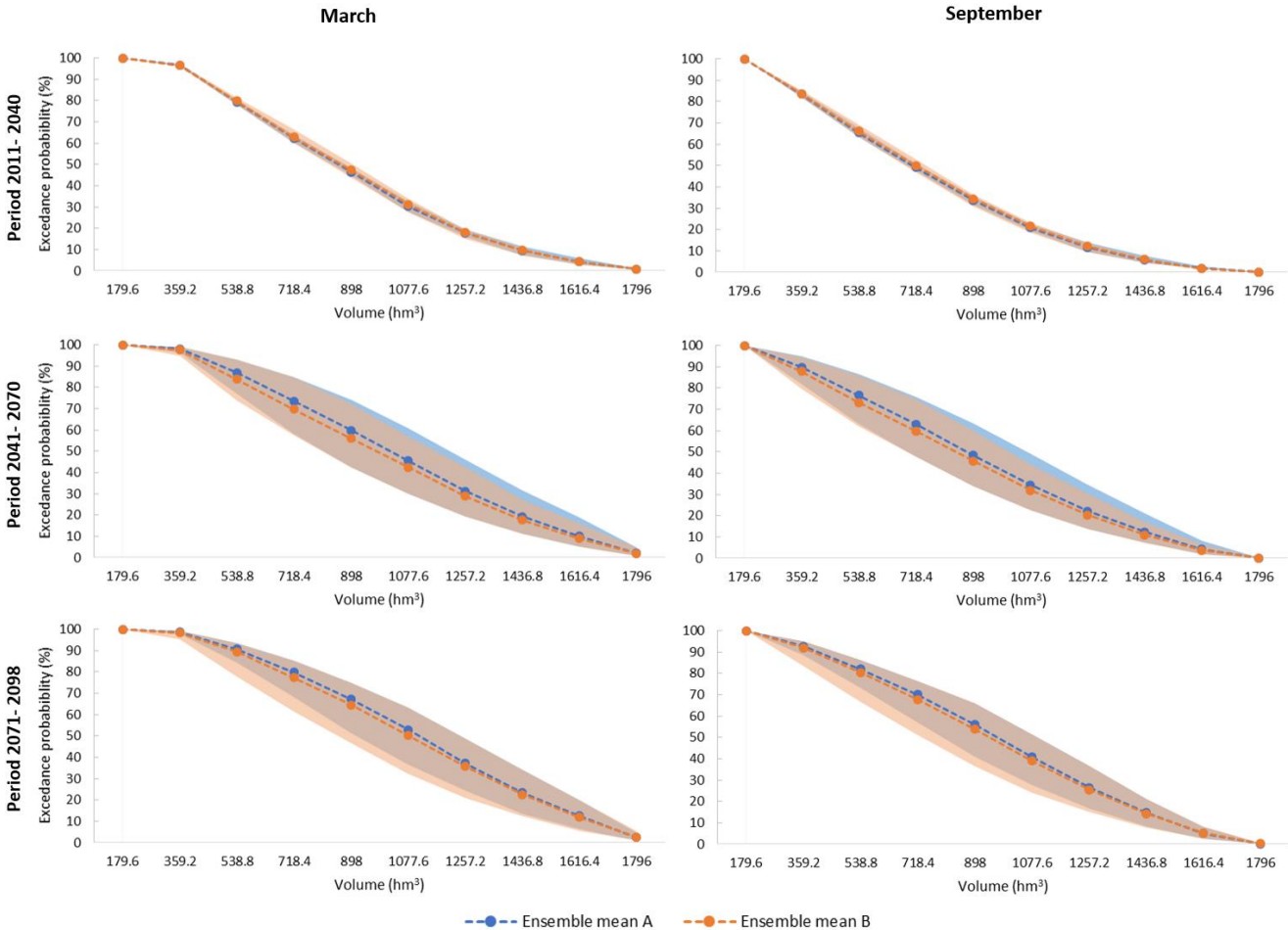

Figure 9. Exceedance probability of the ensembles (shaded areas) coming from options A and B in the start (March) and the end (September) of the irrigation season for the future periods 2011-2040, 2041-2070, and 2071-2098.

In addition, March results show higher percentages of exceedance probability for the same volume if they are compared with those from September. These results are logical due to the winter storage that provides more water resources for the start of the irrigation season, while in September these values are lower due to water allocation during this season and the summer period, which normally lacks precipitation incomes.

For example, in the near future of March, the probabilities of exceeding 50% of total capacity are on average 46% in both

approaches, while in September this value is 34%. Then, these probabilities in the second period of March are 60% (ensemble mean A) and 56% (ensemble mean B), but ranges are between 42%-74% and 42%-72%, respectively. In the same period for September these values are 48% (ensemble mean A) and 46% (ensemble mean B), but ranges are between 34%-63% and 34%-60%, respectively. In the far future the same happens, higher mean values of exceedance probabilities for the same volume and wide ranges covered by the ensemble.

Hence, the dispersion and uncertainty beyond the first period is considerable, as was noted in Fig. 8, and the probabilities of exceeding 50% of total capacity are around 10% higher in March than in September for all periods, indicating more probabilities of water availability in March that may not compromise the irrigation season.

## 5. Discussion

This work has highlighted the most relevant points to be considered for integrating climate projections into decision-making

processes. The proposed methodology is easy to understand and to replicate but it has to be adapted to the features of the case study, so a high level of knowledge of the WRS is an important requirement to implement it. In this case, it was adapted to a Mediterranean basin with water scarcity problems and long periods of drought. Consequently, the more attention we pay to each step, the better the results. In spite of this, the indicator did not provide conclusive results due to the great dispersion of climatic projections, especially in the last two future periods. Therefore, it seems necessary to discuss the process step by

step to estimate possible mistakes and improvements.

First, the data from SWICCA were selected due to the pre-processing they made of filtering the models that best fit in the European area. Despite this, it is stated in the literature that for the Mediterranean area it is very difficult to find reliable data or with enough skill to work with them with confidence (Barranco et al., 2018; Collados-Lara et al., 2018), especially if these are hydrological data (Suárez-Almiñana et al., 2017). This is why we decided to work with meteorological variables,

even though the process may be simpler and shorter using hydrological variables. In Suárez-Almiñana et al. (2017) it was stated that pan-European models do not have yet the capacity of representing the hydrologic characteristics of complex basins. This may be due to the wide-scale of the European hydrological models, where the tight relationship between rivers and aquifers coupled with the high anthropization of rivers (typical of dry areas) is not well represented unless the hydrological model was well tailored to the basin. In addition, it is also important to consider that final results will depend on

the input data selected, so this first step may be the key for the rest of the process. In this way, the proposed methodology would be used in other basins incorporating meteorological variables to avoid this problem.

On the other hand, we believe that the reduction of the reference period is a good choice to start with data more in line with the current situation of the basin. This fact has also been demonstrated in Suárez-Almiñana et al. (2020), where the uncertainty about the effects of climate change on the future inflows of this basin was minimized.

Then, looking at Fig. 4 and Fig. 5, where raw and corrected precipitation and inflows are shown, there is no doubt that the application of some kind of bias correction was necessary. Working with the raw data would lead to unfavourable results for the future, since the underestimation of flows in the headwaters (where the major reservoirs are located) are notable, this fact may also lead to alarming conclusions about the future hydrology in this basin, which may not be correct. Therefore, the quantile mapping technique was applied for both options A and B. This technique is highly recommended in the literature

(Grillakis et al., 2017; Collados-Lara et al., 2018; Manne et al., 2017; Teutschbein and Seibert, 2012), but after having tried other simpler techniques such as month-specific correction factors (Suárez-Almiñana et al., 2017), the differences between their performances are not significant, although the fitting was improved especially in the annual average. It seems that the currently available methods of bias correction may not provide fully satisfactory results, neither a satisfactory physical justification, since they may hide uncertainty rather than reduce it (Ehret et al., 2012).

The combination of NSE and PBIAS statistics also showed how the bias correction did not improve much more the goodness of fit of the ensemble, despite the good calibration of the hydrological model. In fact, they have to be used with caution because PBIAS may be influenced by the uncertainty (Moriasi et al., 2007) and the rating values recommended for the NSE may be too restrictive, since only negative values of NSE indicate an inacceptable performance (Moriasi et al., 2007) and this did not happen in the case of Molinar, Tous, and Sueca when the HBV was tested with historical data, even though they

were very low ($\approx 0.2$). The hydrological model is another source of uncertainty and it has to be considered (Muerth et al., 2013), but it is significantly less important than that provided by the RCMs (Vetter et al., 2014).

All these suggest that the skill of climate change projections needs to be improved in order to work with them effectively. Based on Ehret et al., (2012) this would be achieved by increasing the RCMs resolutions at the convection-permitting scale in combination with ensemble predictions based on sophisticated approaches for ensemble perturbation.

Meanwhile, a future consideration might be the application of improved bias correction methods (Switanek et al., 2017) or a seasonal correction, which may be more relevant for water management and especially in this area, totally conditioned by the irrigation season. However, some authors said that in some cases, the RCMs are not able to reproduce drought statistics from the observed series (Collados-Lara et al., 2018; Cook et al., 2008; Seager et al., 2008), so a correction focussed on drought statistics is also a feasible solution to try to leave out the mismatches between reference periods.

Regarding the impacts on future inflows, they experimented decreases in both options, which is consistent with several studies conducted in this area (Barranco et al., 2018; CEDEX, 2017; Marcos-Garcia et al., 2017). But the behaviour of Tous sub-basin is remarkable because the rate increases until the second period. As mentioned above, this may be conditioned by its connection with the aquifer and the increase in contributions observed in recent years (Hernández Bedolla et al., 2019). This increase in contributions seems to be captured by the models, since the rainfall rate also increases in the first period,

maintaining the average of the baseline until the second period and sinking in the last period. This increase in rainfall

combined with the increasing contributions from the groundwater (included in the hydrological model) and the low water resources of the baseline may lead to those increments in percentage. In any case, the variability of changes between sub-basins is not an isolated case (Folton et al., 2019).

However, if we focus on the average change rates of the whole JRB (Fig. 6, bottom), their values may seem rather low when they are compared to the benchmark study of the CEDEX (2017). This study estimates average reductions (RCPs 4.5 and 8.5) of -7% (near future), -18% (medium future) and -28% (far future) for the entire JRBD, although it is indicated that change rates can be applied to all its points (Barranco et al., 2018). The main reasons for these differences may lie in the reference period of the report (1960-2000) and the lack of bias correction, even though precipitation on the Mediterranean side was underestimated (Barranco et al., 2018). In that reference period, the data before the 80s provides a much more favourable scenario in terms of the availability of water resources compared to the current one. Therefore, when future change rates are obtained, the decreases for the future are more drastic. These simple premises may explain why the change rates of this work are lower or more "optimistic" than those provided by the CEDEX (2017) report.

Then, it was decided to continue with the statistical characteristics of future flows to obtain the drought risk indicators, where the decreasing behaviour observed in the inflows was not equally evident (Fig. 8). Only in the first period a complicated scenario in which the probability of being below 50% of the total storage capacity of the system are 80% can be seen. However, in the rest of the periods the probabilities of being in any of the intervals is practically the same (≈10%). The reason for this is most clearly seen in the probabilities of exceedance capacity (Fig. 9), where the range of probabilities covered by the ensemble is very wide, indicating that their dispersion from the second period onwards is very high and no conclusions can be drawn from them.

The results from the simulation of the future water management supports the dispersion theories extracted from the evaluation of the indicators and the exceedance probabilities, since in Fig. 7 the ensemble is occupying practically the entire storage volume of the WRS in both options (larger in option B), indicating that anything could happen and confirming that the uncertainty of climate projections is considerable. In addition, looking at Fig. 7, it seems that the bias correction of flows provide more dispersion and also lower average values of water storage, which from the point of view of water management is more interesting since the worst scenarios were considered, but the uncertainty is so high that any option can be chosen. In this way, we can understand why it is better to work in terms of probabilities when the future is so uncertain.

Furthermore, the fact of choosing the dammed volumes and their evolution as a reference is motivated by the great influence that these volumes have on the JRB drought indicator (CHJ, 2018), representing almost 50% of the indicator's value (Haro-Monteagudo et al., 2017). Therefore, the proposed indicator can serve as an approximation of the current drought indicator and complement it.

Although the results are not conclusive, the proposed methodology is feasible when integrating future projections in the decision-making processes, but for this area the skill of climate projections needs to be improved. This uncertainty and the absence of a clear and real danger leads the decision-makers to justify inaction (Lemos and Rood, 2010), but the decreasing tendencies of future flows and the indicator for the near future are signals to be considered, since taking preventive measures

may be the key to avoid severe socioeconomic and environmental impacts. In addition, this type of study seeks to complement or improve the RBMP, but at the same time, its conclusions affect the delicate balance of the system, highlighting the need to review the current operating rules for the future, as well as the water allocations and other related elements of the system.

## 6. Conclusion

In this paper, a robust and adaptive methodology was presented to support the decision-making process in complex basins, taking into account the influence of climate change in WRPM. The new perspective of this method regarding current approaches lies in the integration of climate change projections into a model chain to perform future management and drought risk assessments, with an emphasis on improving the process with the characterisation of natural inflows. This approach introduces an important advantage trying to fit climate data to the WRS through some adjustment and bias

correction processes, which are essential to adapt climate data and models as much as possible to the basin features.

All the process was designed with the objective in mind of transforming the information provided by climate services into useful information for decision-making, in order to be understood and trusted by stakeholders and decision-makers. Hence, the key outcomes that can be extracted at different points of the model chain (future change rates, water storage, and drought risk indicator) are presented in intuitive formats to be easily understood. In this way, it is expected that the existing gap

between climate services and WRPM decision-making will be reduced, contributing to a better adaptation to climate change.

The application of this methodology to the JRB has shown how it can be tailored to systems affected by high hydrologic variability and recurrent droughts, taking into account that a good knowledge of the WRS features is essential to get good results. In this case, after the adjustment of the reference period to incorporate an abrupt decrease in average precipitation ("80s effect") and the application of both types of bias correction (to meteorological and hydrological variables), a

concerning decrease of future inflows was observed. These decreasing rates were also reflected in the drought risk indicator for the near future, where the very high probability of having values of the total water stored in the WRS less than half of the total storage capacity calls for action.

Unfortunately, the results from the middle century onwards are not conclusive due to the high dispersion of the EMs, indicating that there is a much higher uncertainty in predicting the future more than 30 years in advance. This leads to the

conclusion that the skill of climate projections needs to be improved to overcome the difficulties to extract robust and reliable results from them. In this way, another branch of the above-mentioned gap could be reduced. Despite this, the improved methodology constitutes a step forward in the inclusion of climate projections in the WRPM decision-making process. And for the JRB case of study, results obtained show that it is time for action to mitigate the impacts in the near future.

## 7. Data availability

The full Spain02 v4 dataset is freely distributed (in NetCDF format) for research purposes (http://www.meteo.unican.es/files/images/copyright_en.pdf) from the Escenarios-PNACC dataset from the UC climate data service. It is available at http://www.meteo.unican.es/datasets/spain02.

The climate projection from SWICCA portal can be freely downloaded at http://swicca.climate.copernicus.eu/indicator-interface/graphs-and-download/ under Creative Commons Attribution-ShareAlike 4.0 International (CC BY-SA 4.0) license conditions.

The natural flows from the Júcar River Basin were provided by the JRBA for research purposes.

## 8. Author contribution

SSA, AS and JM collected the data. SSA, AS, JPA and JA designed the methodology. SSA performed the calculations and analysed the results with AS and JA. SSA prepared the manuscript with contributions from all co-authors: JPA, JM, JA, and AS.

## 9. Competing interests

The authors declare that they have no conflict of interest.

## 10. Acknowledgments

The authors thank the Spanish Research Agency (MINECO) for the financial support to ERAS project (CTM2016-77804-P, including EU-FEDER funds). Additionally, we also value the support provided by the European Community's in financing the projects SWICCA (ECMRWF-Copernicus-FA 2015/C3S_441-LOT1/SMHI) and IMPREX (H2020-WATER-2014-2015, 641811).

It is also important to mention the Research and Development Support Programme (PAID-01-17) from the Universitat Politècnica de València for encouraging and facilitating training contracts for research staff.

Finally, the authors thank AEMET and UC for the data provided for this work (Spain02 v4 dataset, available at http://www.meteo.unican.es/datasets/spain02).

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
