# Peer review of "Risk assessment in water resources planning under climate change at the Júcar River Basin"

_Hydrology and Earth System Sciences, 2019_

## Referee Comment (RC1) · Anonymous Referee #1 · 22 Nov 2019

This manuscript introduces a methodology to assess the effect of climate change on water resources systems by using a model chain connecting climate model results, a semi-distributed rainfall-runoff model and a water management simulation model. While I think the themes dealt with in this work are relevant and interesting for the scientific community, I am afraid the manuscript requires a severe degree of revision before it can be considered for publication in HESS. To be accepted for publication in any journal, the first thing the authors should do is a thorough language and structure revision. At present, the text is clumsy and difficult to read. In addition, I found some

relevant information out of place, e.g., the mention to the improvement introduced in the study in page 4, which should appear in the conclusions section too (I will come back to the added value of the study later), or the description of the drought risk indicator calculation in section 4.3 that should be in methods (I will come back to this too). Regarding the content of the manuscript, my first major concern is its absolute lack of focus. Section 1 introduces a series of concepts mostly disconnected from each other and somehow irrelevant for the rest of the manuscript. Half of the introduction talks about climate services and how to deal with the data they provide, yet no further mention is made to them later in the text either in the discussion of the conclusions section. The last 20 lines of the introduction more or less describe the problem the authors want to study and one could discern what the research objective is. However, it is not the task of the reader to guess the objectives of research work. The authors must explicit what they want to achieve with their work and communicate this to the reader in an efficient and straightforward way. I would like to add that developing a methodology is not an objective in itself but rather a tool to pursue the answer to a research question. Concluding that the methodology is general enough to be applied in other case studies would be an acceptable conclusion though. I do not have major concerns about the methodological approach of the research: climate→hydrology→management, but there is an evident need to improve the clarity of exposition in terms of sections arrangement and description of methods. With regard to sections arrangement, I have the impression that sections 2 and 3 are not separated appropriately. I would suggest that the case study was presented first. The Jucar River Basin is an extensively studied catchment in literature, especially from this research group. Still, I think the system deserves having its own section. Afterward, the whole section 2 and subsections 3.1 through to 3.3 should be merged in a single "materials and methods" section. Coming now to the description of methods, I think the part between lines 89 through to 115 requires a better explanation, including justification of figures 1 and 2. Line 89 reads: "In this section, a distinction between the current assessment in the management of water resources and the analysis of risks was made, despite of being intimately related". Disregarding the quality of this sentence, there is nothing in section 2 that actually deals with that distinction unless the reader is imaginative to say the least. My assumption is that the authors call one thing (current way) to management made on the basis of current/past climate analysis and they call another thing (risk analysis way?) to the assessment of management under future climate, and they argue that the two approaches should be integrated. I do not understand the reasons for such extravagant differentiation of a traditional present versus future analysis that does not add anything conceptually new to the current state-of-the-art. Now is when things get spicy. The authors mention that the novelty introduced in this research "lies in the characterization of future natural inflows and the combination of the management and risk assessments". But, what is new about determining streamflow under future climate conditions and compare it against present conditions? I want to think this must be a writing error from the authors as I do see more value to the results they present further than just comparing two situations. Continuing down the line, I think section 3.3 is the core description of the methodology. I suggest it appears earlier in the text and that it uses a more generalized language only mentioning that modules from the Aquatool software will be used (substitute Aquatool modules' names by generic names, e.g., EVALHIDАň→rainfall-runoff model, MASHWIN→stochastic streamflow series generator, SIMGES→water management simulation model). Next, I suggest sections 2.1 and 3.1 are merged into a single 'current and future climate (or better name)' section. Section 3.2 could be a subsection of the new merged section. By the way, the words precipitation and temperature do not appear so often in the text and are not that long to require using an acronym. I suggest you revise this. Finally, all the models that are actually part of the Aquatool software would be better together under an 'Aquatool modeling package (or better name)' section. I would like the authors to clarify their position with regard to bias correction. By the way, figure 3 is an absolute mess, it should be revised for clarity. The authors claim in the discussion that "working with the raw data would lead to unfavorable results for the future since the underestimation of flows in the headwaters is notable, this fact may also lead to alarming conclusions about the

future hydrology in this basin". But, in figure 6, we see that bias correction actually changes one problem for another, especially at the resource generating catchments of Alarcon and Contreras. While the uncorrected data fits visually well precipitation between March and September (dry months), underestimating it during the wet months in the winter, the bias-corrected data overestimates spring and summer precipitation while still underestimating winter precipitation. This is potentially a problem if the extra amount of water in the summer introduced with bias correction exceeds the winter deficit of the uncorrected data. I think this might be explored. The descriptions in 4.2.1 and 4.2.2 correspond to the methods section. The authors should limit to describe the results in these sections. Moreover, I think the section would benefit from merging figures 7 and 8, although I am not sure whether the first column in figure 8 should be maintained for what I will mention next. Figure 9 shows the mean rates of streamflow change for the three future periods with regard to the reference period. Did the authors check the rate of change between reference and future periods of non-corrected flows from option B? I think this might be revealing. Also, the size difference between graph A and graph B in figure 9 should be revised. Regarding the final step of the methodology, relative to water management simulation, I think the authors should justify better the added value of using stochastic modeling when they already have a reasonable amount of data (several 30 years series from various climate models) to perform the statistics relative to water storage. I think the results from using the water management simulation model should appear before the ones from the stochastic simulation and, in any case, both results should be comparable (e.g., calculating the drought indicator, or the exceedance probability in September). The exceedance probability at the beginning of the irrigation season might be a relevant result too. Anyway, the results of this section show how the relevance of the bias correction is dampened through the modeling chain. Considering limitations and additional uncertainty introduced by bias correction highlighted by Ehret et al. (2013) and the results of studies like Muerth et al. (2013) who argue about the utility of bias correction in model chains, I think the authors lost a good opportunity to contribute to the existing debate on the added value of bias

correction in the modeling of climate change impacts on water resources. I would not like to finalize my review mentioning that the authors make the wrong use of the term tendency throughout the whole text to my understanding. Mostly because the authors do not show whether their results really follow any trend and whether this is significant.
* * *

---

## Referee Comment (RC2) · Anonymous Referee #2 · 25 Nov 2019

First, and from the point of view of the structure of the paper, its title is too long and inaccurate. The work contains, in addition to the methodology, a case study and results obtained after applying the developed methodology. Second, there is confusion between sections 2, Material and methods, and 3, Case study, since subsections 3.1 and 3.2, and perhaps 3.3, would be better classified as Material and methods. As for figures, figure 1 does not seem necessary and figure 2 is difficult to understand. The introduction lacks the reference to similar works that have incorporated climate change projections in decision-making processes in other basins, not just those in the Mediter-

ranean environment. And this is important since the results of the work show a great dispersion (see figure 12). The Material and methods section is quite robust since this work group has implemented numerous modules, already contrasted, in the Aquatool Decision Support System and now used (hydrological model; management model; water allocation model; stochastic model and risk assessment model) This paper provides the integration of climate projections into the model and its impact on future flows in the basin and on the storage of water in the system. In this sense, it uses nine Ensemble members (table 1) that cause a great dispersion of results, as already mentioned, and an inaccuracy in the conclusions. Would it be possible to use only those that have given better results in the Mediterranean region? On the one hand, they work with flow data in the basin between 1980-2012 and, on the other hand, the reference period is reduced to the 1980-2000 period. However, as can be seen in Figure 5, there are differences in the average year inflows between the different periods. Can the use of these different periods have an influence on the results obtained? The results obtained in figures 6, 7 and 8 are only visually compared. In the text it is written, for example, (lines 353-354): "There can be seen how both HBV models results are generally close to the observed flow values". Would it be possible to specify, from a statistical point of view, the term "close"? The results of figure 9 show a great variability between options A and B, mainly in the two head reservoir, Alarcon and Contreras. In view of the results in Figure 12, could one option be recommended over another? Some minor comments would be: - Figure 2: the acronyms of P and T have not been previously defined - Line 133: the acronym RCM is defined later (see line 236) - Lines 223-226: There are several references to geographical names such as the Albufera of Valencia that are not shown on the map in Figure 4

---

## Author Comment (AC1) · 20 Jan 2020

**Referee #1:**

This manuscript introduces a methodology to assess the effect of climate change on water resources systems by using a model chain connecting climate model results, a semi-distributed rainfall-runoff model and a water management simulation model. While I think the themes dealt with in this work are relevant and interesting for the scientific community, I am afraid the manuscript requires a severe degree of revision before it can be considered for publication in HESS.

As you consider this work is interesting for the scientific community, we will carry out an in-depth review of the manuscript in order to solve all its weak points, both linguistic and structural issues, as the clarification of results.

To be accepted for publication in any journal, the first thing the authors should do is a thorough language and structure revision. At present, the text is clumsy and difficult to read.

We consider that the manuscript may be difficult to read since English is not our mother tongue and the process of writing in detail all the steps of the methodology is very complicated. We understand that some expressions or meanings that we think are correct may not be right and difficult to understand for other readers. For this reason we will make an in-depth language review to make it more understandable and to ensure that grammatical structures and the vocabulary used are correct.

In any case, one of the strengths of the HESS journal is the English language copy-editing, so if after our review of structure, grammar, etc., it needs to be improved, we are sure that this problem will be solved before its publication.

In addition, I found some relevant information out of place, e.g., the mention to the improvement introduced in the study in page 4, which should appear in the conclusions section too (I will come back to the added value of the study later), or the description of the drought risk indicator calculation in section 4.3 that should be in methods (I will come back to this too).

We will respond in more detail later, as you will come back to them in other comments.

Anyway, in the first case you refer to this statement (page 4): "The improvement developed in this study lies in the characterization of future natural inflows and the combination of the management and risk assessments." You are right, we will integrate the main improvement in the conclusions as there we focused in the results obtained in each part and what conclusion we extract from them.

Regarding the description of the drought risk indicator in section 4.3 (Results section), we know it is out of the place, but we considered important to guide the reader in the process in order to know where this results come from, as this is a complicated process and no easy to understand. As we said before, we will come back to this later.

Regarding the content of the manuscript, my first major concern is its absolute lack of focus.

Perhaps we did not guide the text to the key points or results, as there are many points to be covered. As you saw, the study is very complete with different options and we did not want to leave any point out or unexplained. We will solve this problem in the following comments and their responses.

Section 1 introduces a series of concepts mostly disconnected from each other and somehow irrelevant for the rest of the manuscript. Half of the introduction talks about climate services and how to deal with the data they provide, yet no further mention is made to them later in the text either in the discussion of the conclusions section.

We consider that highlight the increasing amount of climate services and the huge available data to work with is important, as well as the lack of a clear rule for handling them. As we say in the introduction, some authors choose to use the ensemble, others select only those that fit with the observed data from the reference period, then they correct them or not, etc. With this content, our idea was to communicate to the reader the complexity of working with climate change projections. Not only due to their large number, but also due to the amount of work involved in their selection and subsequent treatment, including the inconvenience that in Mediterranean areas the skill of these data may not be sufficient (Suárez-Almiñana et al., 2017; Barranco et al., 2018; Collados-Lara et al., 2018), thus obtaining disperse results with great uncertainty, as it happens in our case.

Furthermore, this information is used to justify the decisions taken during the first steps of the methodology, such as in the choice of SWICCA data as inputs (due to the easy downloading, handling and confidence in the selection of models adapted to all Europe made by SMHI, an institute of recognized prestige in these issues). In addition to the need to correct the data and use all the members of the ensemble for the study. All this was later named or justified in section 2 (lines 128-134), sections 3.1 to 3.3, section 4 (lines 304-308) and section 5 (lines 483-495, lines 504-506) of the manuscript.

Therefore, we do not believe that everything introduced here about climate services and the way to work with them is irrelevant. It is, after all, a way of exposing what we use and why in this study.

However, we will improve the introduction as discussed below.

The last 20 lines of the introduction more or less describe the problem the authors want to study and one could discern what the research objective is. However, it is not the task of the reader to guess the objectives of research work. The authors must explicit what they want to achieve with their work and communicate this to the reader in an efficient and straightforward way.

Based on the previous comment and this one, we realize that we have not reached the objective we wanted explaining all this in the introduction, therefore, the reformulation of the introduction using a clearer and more direct language it is necessary. There, we will indicate why we present all this information, making more references to it in the other sections of the

manuscript and specifying what is the main objective of this methodology developed to help decision makers to face an uncertain future.

I would like to add that developing a methodology is not an objective in itself but rather a tool to pursue the answer to a research question. Concluding that the methodology is general enough to be applied in other case studies would be an acceptable conclusion though.

Thank you for pointing this out. We believe that specifying the question to which we want to give an answer with this methodology is necessary. In this case, the aim is offering a tool to the decision makers that provides both, deterministic and probabilistic intuitive results, for different future periods and thus have the opportunity to apply measures, in order to avoid or mitigate the possible adverse effects of climate change. Using this tool it is also possible to test whether the proposed measures provide an improvement in the future state of the system or not. This is possible by modifying the conditions of the system when applying the measures and re-simulating the risk assessment model to obtain results about greater or lesser probabilities of having certain deficits in the demands, or certain volumes of water resources in the basin.

All this information clarifies why the general methodology was developed and its applicability to other case studies, even though the step of applying the measures and testing them is beyond our scope in this work.

I do not have major concerns about the methodological approach of the research: climate-hydrology-management, but there is an evident need to improve the clarity of exposition in terms of sections arrangement and description of methods.

You are right, it seems that the problem of this manuscript is the way of telling the whole process, which is not clear and a little unstructured, so we will make a great effort to change all these aspects. We detail them below.

With regard to sections arrangement, I have the impression that sections 2 and 3 are not separated appropriately. I would suggest that the case study was presented first. The Jucar River Basin is an extensively studied catchment in literature, especially from this research group. Still, I think the system deserves having its own section. Afterward, the whole section 2 and subsections 3.1 through to 3.3 should be merged in a single "materials and methods" section.

Perfect, we will follow your suggestion, in this way we give more importance to the case study and then we can develop further certain points mentioned above by putting together section 2 and 3.1 to 3.3. This change will allow us to reduce the length of the manuscript and be clearer and more concise by explaining everything in the same place. Despite this restructuring, we also have to clarify that this is a general methodology that has to be adapted to each case study, in this case it was applied to the Jucar River Basin.

Coming now to the description of methods, I think the part between lines 89 through to 115 requires a better explanation, including justification of figures 1 and 2. Line 89 reads: "In this section, a distinction between the current assessment in the management of water resources and the analysis of risks was made, despite of being intimately related". Disregarding the quality of this sentence, there is nothing in section 2 that actually deals with that distinction unless the reader is imaginative to say the least. My assumption is that the authors call one thing (current way) to management made on the basis of current/past climate analysis and they call another thing (risk analysis way?) to the assessment of management under future climate, and they argue that the two approaches should be integrated. I do not understand the reasons for such extravagant differentiation of a traditional present versus future analysis that does not add anything conceptually new to the current state-of-the-art.

We made this differentiation by the way water resources are managed in the case study. Currently, the water allocation or management model is used for water resources planning for a horizon of 6 to 18 years, and then, for real time management and droughts events the managers use the risk assessment, which is normally used for a horizon of 1 to 12 or 24 months. All this is stated in the Júcar River Basin District Management Plan (CHJ, 2015) and the Drought Management Plan (CHJ, 2018).

What we are trying to say with these figures is that we can take advantage of both methods for the future by inserting climate change projections into them and this methodology has not yet been integrated into the River Basin Management Plans design. Thus, Figure 2 shows the steps to do this process and throughout the text, we explain it step by step. At the end, we can extract the results of the evolution of the basin's water resources in a deterministic (Figure 12) and probabilistic (Figure 10) ways to help decision-makers to take decisions for the future.

However, we consider that we did not introduce this information in a clear and understandable way, so we will remove Figure 1 and then we can introduce this information in the introduction section or in the new section dedicated to the case study (commented on later), as it is the background or the current state-of-the-art of this area.

Now is when things get spicy. The authors mention that the novelty introduced in this research "lies in the characterization of future natural inflows and the combination of the management and risk assessments". But, what is new about determining streamflow under future climate conditions and compare it against present conditions? I want to think this must be a writing error from the authors as I do see more value to the results they present further than just comparing two situations.

The point here is the process related to the characterisation of the future river flows and the planning and risk assessments. With this statement, we mean all the process involved in the treatment of climate change projections in order to adapt them to the basin features and then the model chain developed to extract some results that complement each other. All this means the bias correction (before or after the hydrological model), the future simulation of the water management model, the integration of the statistical properties of each ensemble member into the stochastic model to generate multiple and equiprobable series, their integration in the risk assessment model, and finally the extraction of the risk indicator and the evolution of water resources of the basin in the future. We do not refer only to the change rates, which we know are very common in this type of studies. Anyway, as we said in the discussion section (lines 517-526), this change rates for the entire basin may be more reliable

than those from other studies because we used a reference period adapted to the current situation of the basin. This is another new income that has to be considered as an improvement of the technique.

We will include all this information in the text.

Continuing down the line, I think section 3.3 is the core description of the methodology. I suggest it appears earlier in the text and that it uses a more generalized language only mentioning that modules from the Aquatool software will be used (substitute Aquatool modules' names by generic names, e.g., EVALHID rainfall-runoff model, MASHWIN stochastic streamflow series generator, SIMGES water management simulation model).

Yes, this section corresponds with Figure 2 and Figure 3, and there is where the main part of the methodology is detailed. In this case, as we agreed to join sections 2 and 31-3.3, this part may be the core of that material and methods section with the help of Figure 2 and a new version of Figure 3, as you demand below. Of course, we can use a more generalized language substituting the names of the Aquatool modules by the name of the models. Then, we will add a new section to introduce this software and its modules as you suggested below.

Next, I suggest sections 2.1 and 3.1 are merged into a single 'current and future climate (or better name)' section. Section 3.2 could be a subsection of the new merged section.

We agree with you, the parts of sections 2 and 3 with the same information will be merged in single sections, as we said before.

By the way, the words precipitation and temperature do not appear so often in the text and are not that long to require using an acronym. I suggest you revise this.

We agree with you, so we will remove the acronyms and we will write precipitation and temperature throughout the text and figures.

Finally, all the models that are actually part of the Aquatool software would be better together under an 'Aquatool modeling package (or better name)' section.

Yes, that makes sense if we name the Aquatool modules with a more generalized language as we agreed before. The name of the section may be Aquatool Decision Support System Shell or similar, where we will explain in detail the features of this software and the modules used for building the different models.

I would like the authors to clarify their position with regard to bias correction. By the way, figure 3 is an absolute mess, it should be revised for clarity. The authors claim in the discussion that "working with the raw data would lead to unfavorable results for the future since the underestimation of flows in the headwaters is notable, this fact may also lead to alarming conclusions about the future hydrology in this basin". But, in figure 6, we see that bias correction actually changes one problem for another, especially at the resource generating

catchments of Alarcon and Contreras. While the uncorrected data fits visually well precipitation between March and September (dry months), underestimating it during the wet months in the winter, the bias-corrected data overestimates spring and summer precipitation while still underestimating winter precipitation. This is potentially a problem if the extra amount of water in the summer introduced with bias correction exceeds the winter deficit of the uncorrected data. I think this might be explored.

We thought that Figure 3 was a good way to introduce the reader to the two options considered in the characterization of future natural flows. There are all the steps we followed in both options and we think that explaining them belter in the text will be enough to understand it properly. However, we can remove it or make it simpler to the reader, as we have to restructure the manuscript, we will think about it.

Regarding the bias correction, looking at Figure 8 (Non corrected flows) we decided that it was necessary to correct the data because the underestimation of flows in the Alarcon and Contreras basins was huge. This is a problem from the point of view of water management as there is where the main reservoirs of the system are placed. This means that if we use this data, we are accepting that the resources we are taking as inputs are much lower than those that are really there, which is not acceptable in this field.

Thus, once we decided to correct the data, the most recommended method to do it was the quantile mapping, which tries to keep the mean and standard deviation of the reference series (Collados-Lara et al., 2018). Then, we applied it but the results were not convincing because precipitations and flows were overestimated in spring and summer months. Despite this, we accepted them as the differences between the averages were minimised and the flows of the headwaters basins were better fitted to the observed values (Figure 7).

However, we tried other techniques for bias correction in previous studies, as month-specific correction factors (Suárez-Almiñana et al., 2017), which results were not so different from those of this study. Hence, we think that the currently available methods of bias correction may not provide satisfactory fittings to the observed series of this area. Despite this, a future consideration can be the application of seasonal corrections or one method that be capable of catch the drought statistics of the observed data, as we know that the RCMs are not able to reproduce them in some cases (Collados-Lara et al., 2018; Cook et al., 2008; Seager et al., 2008). We referred to all this in the discussion section, lines 491-499.

Anyway, in the next comments we refer to the bias correction issue in more detail.

The descriptions in 4.2.1 and 4.2.2 correspond to the methods section. The authors should limit to describe the results in these sections.

We thought that repeating the process in these sections was necessary to guide the reader through them and refer to how they were obtained. Following the whole process without any guidance may be complicated and can lead the reader into confusion.

Anyway, when we restructure the manuscript we will consider removing or reducing these descriptions from these sections, as this is not the place for them.

Moreover, I think the section would benefit from merging figures 7 and 8, although I am not sure whether the first column in figure 8 should be maintained for what I will mention next.

We are not sure about putting these two figures together as they may be too stacked, the text may be illegible. However, we will consider their combination when we restructure the text.

Figure 9 shows the mean rates of streamflow change for the three future periods with regard to the reference period. Did the authors check the rate of change between reference and future periods of non-corrected flows from option B? I think this might be revealing. Also, the size difference between graph A and graph B in figure 9 should be revised.

We did not check the average change rate of non-corrected flows because the flows of the reference period are hugely underestimated in the reference period compared to the observed data (Figure 8), mostly in the headwaters basins.

Anyway, the change rates of river flow are available in the SWICCA portal for each future period, but the results without skill in the re-forecast analysis are almost in the whole basin. Thus, we thought that using these change rates would not give us realistic results for the future. However, we can test it with our flow data, and if they are revealing we will include them in the study.

We will review and fix the size differences in Figure 9.

Regarding the final step of the methodology, relative to water management simulation, I think the authors should justify better the added value of using stochastic modeling when they already have a reasonable amount of data (several 30 years series from various climate models) to perform the statistics relative to water storage.

We think that only 9 series (one for each ensemble member) are not enough to apply the risk assessment process, even if we divided it in periods of 30 years. The more equiprobable series we generate with the statistical properties of each ensemble member, more agreement between them, which means more reliable results. This statement is not shown in the ensemble risk indicator (Figure 10) as the ensemble members are quite disperse (dry or wet periods) and they complement each other in the ensemble resulting in almost the same probabilities of being in any volume interval of the total capacity of the system.

I think the results from using the water management simulation model should appear before the ones from the stochastic simulation and, in any case, both results should be comparable (e.g., calculating the drought indicator, or the exceedance probability in September).

We can place the results of water management before those of the risk management. Actually, they are already comparable because in both cases we are showing the evolution of the water resources of the system, one in form of risk indicator with probabilities for each future period and in the other with mean volumes of the ensemble and the range covered by it for the entire period. Thus, they complement each other.

The exceedance probability at the beginning of the irrigation season might be a relevant result too.

You are right, in fact it is possible to extract the exceedance probability for each month, but we focused on September because it is the end of the irrigation season and the end of the hydrological year. Thus, this result is probably the one that best summarizes the final state of each campaign. In addition, this is a meaningful data for the stakeholders because it is better understood by them.

Anyway, we will think about including the exceedance probability at the beginning of the irrigation season (March) because it can be a good way of informing the irrigation associations about the possibilities of having shortages and take measures to avoid them. Thus, we may show both results in the reviewed manuscript.

Anyway, the results of this section show how the relevance of the bias correction is dampened through the modeling chain. Considering limitations and additional uncertainty introduced by bias correction highlighted by Ehret et al. (2013) and the results of studies like Muerth et al. (2013) who argue about the utility of bias correction in model chains, I think the authors lost a good opportunity to contribute to the existing debate on the added value of bias correction in the modeling of climate change impacts on water resources.

You are right, the debate on the added value of bias correction in the modelling of climate change impacts on water resources is very interesting, but that is not one of the purposes of this study.

What is important here is that the raw data were very inappropriate in this case, and we applied the most recommended method to correct them, the quantile mapping (QM). We know that it has its pros and cons, so we decided to test it on the meteorological data and on the flows, just in case there was a notable difference between them and be able to recommend the better option. In this case, the difference was not significant.

However, the papers of  Ehret et al. (2012) and Muerth et al. (2013) are very interesting, such as the one by Teutschbein and Seibert (2013) which applies the QM in different seasons and the one by Switanek et al. (2017) in which the QM method for climate change applications is improved. We plan to mention all these works and go a bit further into the subject of bias correction in the discussion, but as you will understand, we cannot develop in detail this part since the important thing in the paper is the methodology and its adaptation to the case study.

By this, we mean that the methodology will already be developed by the time we are able to obtain better climate change projections adapted to each area or correct the data to a better fitting to those observed in every sense.

I would not like to finalize my review mentioning that the authors make the wrong use of the term tendency throughout the whole text to my understanding. Mostly because the authors do not show whether their results really follow any trend and whether this is significant.

In this case, the term tendency refers to the decrease of flows as we approach 2100, which is shown in the average change rate of the Jucar River Basin (Figure 9) and in the average ensemble of the system's resource after applying the water allocation model (Figure 12).

Perhaps the trend is not so evident because the reduction is relatively low (from 1% to 12%), but considering that we are working with the Jucar River system, these decreases can lead to major problems of water deficits and huge economic losses.

Therefore, these small decreases are significant, as the basin is already stressed (demands/resources ≈ 1) and this presents a big challenge for decision makers during extreme events such as droughts, which will be more intense and longer according to authors such as CEDEX, 2017 and Marcos-Garcia et al., 2017.

We will try to provide more information about this topic in order to the reader understand why this small decrease is relevant. In addition, we will not use the word "tendency", we will directly refer to the "decrease" in order to avoid any misunderstanding.

**We hope that our responses to the reviewers' comments and the changes we will make in the manuscript will be enough to be considered for publication in the HESS journal.**

---

## Author Comment (AC2) · 21 Jan 2020

**Referee #2:**

First, and from the point of view of the structure of the paper, its title is too long and inaccurate. The work contains, in addition to the methodology, a case study and results obtained after applying the developed methodology.

In the title we wanted to highlight the importance of the methodology, as it integrates the water planning, the risk assessment and the possibility of making the bias correction in different ways that leads to the characterisation of natural flows. We did not include the case study in the title since it is a methodology that can be developed in other basins, taking into account its features. Thus, we can say that it is a generalist methodology and within its main framework (Fig. 2) decisions to the basin involved are taken. In this case, we chose the Júcar River Basin due to its hydrological features described in section 3, such as the high hydrological variability that lead to recurrent multiannual droughts and the exploitation rate of the water resources ($\approx 90\%$), among others.

However, if you find it convenient, we can give a more concise format to the title integrating also the case study, these are some options we are considering:

- Characterisation of natural flows and modelling chain methodologies for risk assessment in water planning under climate change at Jucar River Basin.
- Methodology for risk assessment in water resource planning under climate change at Jucar River Basin.

Second, there is confusion between sections 2, Material and methods, and 3, Case study, since subsections 3.1 and 3.2, and perhaps 3.3, would be better classified as Material and methods.

You are right, now we think that developing the general methodology in section 2 and then describe the details of the methodology applied to this case study in the subsections of section 3 was a mistake. To resolve this, we will include an individual section to the case study, as it is a very problematic and interesting basin from the point of view of water management. Then, after this new section, we will develop the material and methods by joining section 2 and sub-sections 3.1, 3.2 and 3.3. In this way, we could reduce the length of the manuscript by bringing everything together in the same section, giving more sense to the structure of the document.

As for figures, figure 1 does not seem necessary and figure 2 is difficult to understand.

Figure 1 was introduced as a small clarification that water management and risk assessment are closely related and how each one works, then in Figure 2 we highlighted the key points of each process. However, we believe that removing Figure 1 and its explanation in this section is necessary. We can include something about this relation in the introduction and link it to the objective of the work, as a background for the way of working in this area up to now. Another option is to include this differentiation in the case study section, as it is part of a specialisation related to water resource management in problematic basins such as the Jucar River Basin, which are stressed due to high exploitation rates (demands/resources $\approx 1$).

Regarding Figure 2, it is a general diagram of the methodology, which together with the surrounding text gives a general and quite explicit idea of what the study is about, detailing the main points and how to reach them. We decided to build it in this way in order to not

overburden the reader with too much information in a single figure, as the process can be confusing and tedious for outsiders in this field.

Therefore, Figure 1 will be removed from the manuscript and we will try to redo Figure 2 to make it more understandable, for example by including the hydrological model and the bias correction in the item "Characterization of future natural flows". In the case that the way of doing it does not convince us, we will develop in the text its peculiarities to clarify the key points and guide the reader through the following sections, where each point will be developed in more detail.

The introduction lacks the reference to similar works that have incorporated climate change projections in decision-making processes in other basins, not just those in the Mediterranean environment. And this is important since the results of the work show a great dispersion (see figure 12).

In this part we decided to focus on studies carried out in this area due to the great dispersion of our results. In this way we justify them since most of the authors agree that the skill of climate change projections of this area is very low and usually they are not capable of representing the characteristics of historical droughts (Collados-Lara et al., 2018; Cook et al., 2008; Seager et al., 2008).

However, if we name similar studies developed in other areas, we could highlight the differences in RCM skills and the dispersion of results depending on the geographical area where they are applied. These differences could be named in both the introduction and the discussion.

As an example, we can name the studies developed in Sweden by Teutschbein and Seibert (2013) and in Germany by Hattermann et al. (2014).

The Material and methods section is quite robust since this work group has implemented numerous modules, already contrasted, in the Aquatool Decision Support System and now used (hydrological model; management model; water allocation model; stochastic model and risk assessment model). This paper provides the integration of climate projections into the model and its impact on future flows in the basin and on the storage of water in the system. In this sense, it uses nine Ensemble members (table 1) that cause a great dispersion of results, as already mentioned, and an inaccuracy in the conclusions. Would it be possible to use only those that have given better results in the Mediterranean region?

As we say in the text, we decided to use the ensemble provided by the SWICCA portal (Table 1), where they selected the members that were most suitable for all Europe. However, none of them provides a good enough fit in this area, both uncorrected and corrected, as they are not able to fit perfectly with the observed data and they are not capable of reproducing the statistical characteristics or trends in the average year (Figures 6 and 8) or over the whole period, neither the characteristics of the historical droughts. For this reason we thought that the use of the whole ensemble would provide us with more options and more robustness to the study, considering more options, since increasing the number of ensemble members reduces the sampling uncertainty (Collados-Lara et al., 2018; Thompson et al., 2017).

Despite all the efforts we made in the first part of the methodology to reduce the uncertainty provided by the RCMs, such as shortening the reference period to be more in line with the current situation of the basin or correcting both the meteorological data and the flows, this was not possible. Therefore the results are quite dispersed. This reveals the need to improve the skill of climate projections and the use of more sophisticated bias correction techniques, as we said in the discussion.

On the one hand, they work with flow data in the basin between 1980-2012 and, on the other hand, the reference period is reduced to the 1980-2000 period. However, as can be seen in Figure 5, there are differences in the average year inflows between the different periods. Can the use of these different periods have an influence on the results obtained?

In this basin, it is advisable to work with data from the period 1980-2012 (Reference), since using series with periods prior to 1980 can lead to an overestimation of water resources.This is due to the so-called "effect 80" (Pérez-Martín et al., 2013; Hernández Bedolla et al., 2019), mentioned and explained in the manuscript. It is a significant decrease in rainfall and inflows in the basin from the 1980s onwards. Thus, this is the reason why we decided to shorten the reference period provided by SWICCA (1971-2000) to 1980-2000, in order to try to better represent the current situation of the basin. In this way, we tried to decrease the uncertainty of the magnitude of future changes when we compare future flows with a reference period that represents the basin, such as the results of the average change rates of the whole basin (Figure 9).

Figure 5 shows how the inflows for both periods (1980-2012 and 1980-2000) can be considered equivalent since the difference between their averages is not significant. However, if we had used the period 1971-2000, we would have a different and unrealistic perception of the basin at this time, since water resources are greater than in reality and by correcting future data this would also be transferred to future periods.

Answering your question, yes, the use of the different periods influences the final results, but in this case the difference between using the period 1980-2012 and 1980-2000 is not significant. Indeed, we are in the process of publishing a paper (Suárez-Almiñana et al, *in press*) that compares the change rates for the whole basin (similar to Figure 9) using the three reference periods named before (1980-2012, 1980-2000 and 1971-2000), among others. In that paper we conclude that the average change rates of the basin are very similar when comparing the future flows of each period (2011-2040, 2041-2070, 2071-2098) with the flows of the reference periods 1980-2012 and 1980-2000. However, when those future periods are compared to the reference period 1971-2000, the average change rates are more drastic (up to -23% at the end of the century), which is logical since this period has more resources available, leading to a more extreme and alarmist conclusion than in this case where the average change rates are between -11% and -12% for the whole basin.

Thus, we think that the shortening of the period was a good decision and the differences between the reference periods 1980-2012 and 1980-2000 would not produce very different results.

The results obtained in figures 6, 7 and 8 are only visually compared. In the text it is written, for example, (lines 353-354): "There can be seen how both HBV models results are generally close to the observed flow values". Would it be possible to specify, from a statistical point of view, the term "close"?

In this case, with the term "close" we refer to the visual distance between their averages, but we can provide some table with basic statistics or Goodness-of-fit functions as the Mean Error (ME), the Root Mean Square Error (RMS), the Nash-Sutcliffe efficiency (NSE), etc. to give robustness to this statement.

The results of figure 9 show a great variability between options A and B, mainly in the two head reservoir, Alarcon and Contreras. In view of the results in Figure 12, could one option be recommended over another?

We cannot choose one option over another because the results obtained are not conclusive and very similar from both options. However, Figure 12 shows how the average of the ensemble is lower in option B and the shading area reaches much lower values than in option A. Therefore, despite the fact that option B is more dispersed, if we chose it we would be working from the side of security against future intense drought events, which seems to be more frequent and intense in the future (CEDEX, 2017 and Marcos-Garcia et al., 2017).

Some minor comments would be: - Figure 2: the acronyms of P and T have not been previously defined - Line 133: the acronym RCM is defined later (see line 236) - Lines 223-226: There are several references to geographical names such as the Albufera of Valencia that are not shown on the map in Figure 4.

We will define the names of the acronyms (P, T and RCM) prior to their appearance in the text or figures. Regarding the Albufera de Valencia, we will include it in the map of Figure 4.

**We hope that our responses to the reviewers' comments and the changes we will make in the manuscript will be enough to be considered for publication in the HESS journal.**

---

## Author Response (AR1)

**First of all we would like to thank Referee 1 and Referee 2 for their comments and suggestions, we considered all of them to improve our manuscript. We will respond to the comments point by point, answering them and telling where the changes were integrated in the manuscript.**

**Referee #1:**

This manuscript introduces a methodology to assess the effect of climate change on water resources systems by using a model chain connecting climate model results, a semi-distributed rainfall-runoff model and a water management simulation model. While I think the themes dealt with in this work are relevant and interesting for the scientific community, I am afraid the manuscript requires a severe degree of revision before it can be considered for publication in HESS.

Thank you for the comment. As you considered this work interesting for the scientific community, we carried out an in-depth review of the manuscript to solve all its linguistic and structural weaknesses, as well as clarifying the results.

To be accepted for publication in any journal, the first thing the authors should do is a thorough language and structure revision. At present, the text is clumsy and difficult to read.

We consider that the manuscript was difficult to read as English is not our mother tongue and the process of writing in detail all the steps of the methodology was very complicated. Therefore, we made a linguistic and structural revision to make it more understandable and to ensure that the grammatical structures and the vocabulary used are correct.

In any case, one of the strengths of the HESS journal is the English language copy-editing, so if after our language review it needs further improvement in this sense, we are sure that this problem will be solved before its publication.

In addition, I found some relevant information out of place, e.g., the mention to the improvement introduced in the study in page 4, which should appear in the conclusions section too (I will come back to the added value of the study later), or the description of the drought risk indicator calculation in section 4.3 that should be in methods (I will come back to this too).

We will respond in more detail later, as you come back to this in other comments.

Anyway, in the first case you are referring to this statement (page 4): "The improvement developed in this study lies in the characterization of future natural inflows and the combination of the management and risk assessments." You are right, we followed your suggestion and we included the main improvement in the conclusions as well, see lines 583-588 and 595-597.

Regarding the description of the drought risk indicator in section 4.3 (Results), we delete it from this part and it can only be found in the new Material and methods section (3.2.4 and 3.2.5). Despite this, we considered that a short clarification was needed to guide the reader in the process, since it is complicated and not easy to understand. Thus, only one sentence (see lines 449-450) refers to the process in the new results section of the drought risk indicator (4.4).

As we said before, we will come back to this later.

Regarding the content of the manuscript, my first major concern is its absolute lack of focus.

Perhaps we did not guide the text to the key points or results in the previous version, as there were many points to be covered, but we solved this problem with your help, as you can see below. The main reason for the apparent lack of focus was the complexity of the study, which has many steps with different options and we did not want to leave any point out or unexplained.

Section 1 introduces a series of concepts mostly disconnected from each other and somehow irrelevant for the rest of the manuscript. Half of the introduction talks about climate services and how to deal with the data they provide, yet no further mention is made to them later in the text either in the discussion of the conclusions section.

Thank you for this comment.

We consider that it is important to highlight the increasing amount of climate services and data available to work with, as well as the lack of a clear rule for handling them. As we said in the introduction, some authors choose to use the ensemble, others select only those that fit with the observed data from the reference period, then correct them or not, etc. With this content we wanted to communicate the complexity of working with climate change projections, not only due to their large number, but also due to the amount of work involved in their selection and subsequent treatment, including their inherent uncertainty and the inconvenience that in Mediterranean areas the skill of these data may be not enough to extract reliable results from them. As in our case, where we obtained scattered results with great uncertainty.

Besides, this information was used to justify the decisions taken during the early stages of the methodology, such as the choice of SWICCA data as inputs (due to the easy download, handling, and confidence in the selection of Europe-wide models made by the SMHI, a prestigious climate services innovation institution). In addition to the need to correct or adjust the data and use all the ensemble members for the study.

Therefore, we do not believe that everything introduced here about climate services and how to work with them is irrelevant. It is, after all, a way of presenting what we use and why in this study, which is now related or justified in several parts of the text: sections 3.1 (lines 179-181), 3.1.1 (lines 211-212), 3.1.2 (lines 219-221), section 4 (lines 312-316), section 4.2.2 (lines 384-389), section 5 (lines 500-518, line 531, lines 536-538, lines 548-556) and section 6 (lines 601-604). Additionally, the introduction was improved as discussed below in order to give more sense to all this information.

The last 20 lines of the introduction more or less describe the problem the authors want to study and one could discern what the research objective is. However, it is not the task of the reader to guess the objectives of research work. The authors must explicit what they want to achieve with their work and communicate this to the reader in an efficient and straightforward way.

You are right. Based on the previous comments, we realized that the introduction needed to be reformulated and improved in a clearer and more direct way. Therefore, we did it and now all previous information and some other statements are related to the main objectives of this study.

These objectives can be seen in lines 49-51, 72-77 and 80-86, which are related to the proposal of an improved methodology that includes climate change projections in the water planning and management process to help decision-makers to cope with future extreme events and to solve some problems related to the uncertainty of these projections.

I would like to add that developing a methodology is not an objective in itself but rather a tool to pursue the answer to a research question. Concluding that the methodology is general enough to be applied in other case studies would be an acceptable conclusion though.

Thank you for the clarification and the suggestion.

We introduced this sentence in the introduction section to clarify once again why we developed this methodology (lines 81-83): "For this reason, the main objective of this study is to provide an answer for some of the before mentioned issues, where an adaptive tool is developed to support and help basin managers to cope with future extreme events such as droughts, which may be more frequent and intense in the future."

In this case, the tool provides decision makers with both deterministic and probabilistic intuitive results for different future periods. Therefore, looking at these results, they have the opportunity to implement measures to avoid or mitigate the potential adverse effects of climate change. Moreover, this tool can also be used to check whether or not the proposed measures improve the future state of the system (see lines 262-263). This is possible by modifying the system conditions when applying the measures and re-simulating the risk assessment model to obtain results on the greater or lesser probabilities of having certain deficits in the demands, or certain volumes of water resources in the basin. All this information clarifies why the general methodology was developed and its applicability to other case studies, although the step of applying the measures and testing them is beyond our scope in this study.

In this new version, the idea of the general method applicable to complex basins was clarified throughout the text, from the introduction to the conclusions, where it was also included in lines 594-596 that an in-depth knowledge of the basin is required to do it.

I do not have major concerns about the methodological approach of the research: climate-hydrology-management, but there is an evident need to improve the clarity of exposition in terms of sections arrangement and description of methods.

You are right, it seems that the problem in the previous manuscript was the way of telling the whole process, which was not clear and a bit unstructured, so we made a great effort to change all these aspects. They are detailed below.

With regard to sections arrangement, I have the impression that sections 2 and 3 are not separated appropriately. I would suggest that the case study was presented first. The Jucar River

Basin is an extensively studied catchment in literature, especially from this research group. Still, I think the system deserves having its own section. Afterward, the whole section 2 and subsections 3.1 through to 3.3 should be merged in a single "materials and methods" section.

Thank you for the suggestion. In the new version we presented the case study in its own Section 2, including more information about the current management of this area. Then, we merged all the related parts of the previous sections 2 and 3 into the new Material and methods section to have all this information in the same place and to avoid duplication.

Coming now to the description of methods, I think the part between lines 89 through to 115 requires a better explanation, including justification of figures 1 and 2. Line 89 reads: "In this section, a distinction between the current assessment in the management of water resources and the analysis of risks was made, despite of being intimately related". Disregarding the quality of this sentence, there is nothing in section 2 that actually deals with that distinction unless the reader is imaginative to say the least. My assumption is that the authors call one thing (current way) to management made on the basis of current/past climate analysis and they call another thing (risk analysis way?) to the assessment of management under future climate, and they argue that the two approaches should be integrated. I do not understand the reasons for such extravagant differentiation of a traditional present versus future analysis that does not add anything conceptually new to the current state-of-the-art.

We made this differentiation based on how water resources are managed in this basin. Currently, the water allocation model is used for water resources planning over horizons of 6 to 18 years. Then, for real-time management and drought events, water managers use the risk assessment, which is normally used for a horizon of 1 to 12 or 24 months. All this is reflected in the Júcar River Basin District Management Plan (CHJ, 2015) and the Drought Management Plan (CHJ, 2018), which were named in the case study section.

What we were trying to say with these figures is that we can take advantage of both methods for the future by inserting climate change projections. Furthermore, this methodology has not yet been integrated into the River Basin Management Plans design.

Thus, we decided to remove Figure 1 and its explanation, as they were confusing and added nothing new to this specific case. However, part of this information was commented in the introduction and case study sections (lines 75-81, 132-135).

Then, Figure 2 in combination with Figure 3 showed the steps of the methodology to extract the risk and management results. As they were confusing and a bit difficult to understand, we merged both figures in the new Figure 2, which is now clearer and more understandable. This figure combined with the description in the text are ensuring a complete understanding of the process.

Now is when things get spicy. The authors mention that the novelty introduced in this research "lies in the characterization of future natural inflows and the combination of the management and risk assessments". But, what is new about determining streamflow under future climate conditions and compare it against present conditions? I want to think this must be a writing error from the authors as I do see more value to the results they present further than just comparing two situations.

The point here is the process related to the characterisation of the future river flows and the planning and risk assessments. With this statement, we refer to the whole process involved in the treatment of climate change projections to adapt them to the basin features and then to the modelling chain designed to extract some results that complement each other.

All this was reformulated from the introduction and explained in detail in the Material and methods section, since we are not referring only to the change rates, which we know are very common in this type of studies. However, as we said in the discussion section (lines 548-556), these change rates for the entire basin may be more reliable than those from other studies because we used a reference period adapted to the current situation of the basin. This is another new income that has to be considered as a technical improvement.

Continuing down the line, I think section 3.3 is the core description of the methodology. I suggest it appears earlier in the text and that it uses a more generalized language only mentioning that modules from the Aquatool software will be used (substitute Aquatool modules' names by generic names, e.g., EVALHID rainfall-runoff model, MASHWIN stochastic streamflow series generator, SIMGES water management simulation model).

Thank you for the suggestion. Section 3.3 is where the main part of the methodology was detailed together with Figures 2 and 3. In the new version of the manuscript, this part is the core of the Material and methods section with the help of the new Figure 2. Then, this section was completed in the subsections where the methodology was adapted to the basin. In addition, we replaced the names of the Aquatool modules by the general name of the models and a new section was included to introduce this software and its modules, as you suggested below.

Next, I suggest sections 2.1 and 3.1 are merged into a single 'current and future climate (or better name)' section. Section 3.2 could be a subsection of the new merged section.

We agree, sections 2.1 and 3.1 were merged in the new section 3.1 called "Climate change projections and historical local data" and previous section 3.2 is now a subsection of the new merged section, as you suggested.

By the way, the words precipitation and temperature do not appear so often in the text and are not that long to require using an acronym. I suggest you revise this.

Thank you for the suggestion. We remove the acronyms from the text and figures.

Finally, all the models that are actually part of the Aquatool software would be better together under an 'Aquatool modeling package (or better name)' section.

You are right. In the new section 3.2.1, called "AQUATOOL Decision Support System Shell (DSSS)", we explained in detail the features of this software and the modules used for building the different models. In the rest of the text we used a more generalized language for the models, as we said before.

I would like the authors to clarify their position with regard to bias correction. By the way, figure 3 is an absolute mess, it should be revised for clarity. The authors claim in the discussion that "working with the raw data would lead to unfavorable results for the future since the underestimation of flows in the headwaters is notable, this fact may also lead to alarming conclusions about the future hydrology in this basin". But, in figure 6, we see that bias correction actually changes one problem for another, especially at the resource generating catchments of Alarcon and Contreras. While the uncorrected data fits visually well precipitation between March and September (dry months), underestimating it during the wet months in the winter, the bias-corrected data overestimates spring and summer precipitation while still underestimating winter precipitation. This is potentially a problem if the extra amount of water in the summer introduced with bias correction exceeds the winter deficit of the uncorrected data. I think this might be explored.

We thought that Figure 3 was a good way to present the two options considered in the characterization of natural flows, but it seems that this figure was difficult to understand and we decided to merge Figures 2 and 3 in the new Figure 2 to show all the methodology in the same figure. Thus, this part was more understandable and its description was completed in the text, as mentioned before.

Regarding the bias correction, as we said in section 4.2.2 and lines 514-517 (discussion section), we decided that a bias correction was needed due to the huge underestimation of flows in the Alarcon and Contreras sub-basins, which is a problem from the point of view of water management since there are placed the main reservoirs of the system. This means that if we use this data, we are accepting that the resources we are taking as inputs are much lower than those that actually exist, which is not acceptable in this field.

Thus, we applied the most recommended method in the literature, but the results of the bias correction were not convincing because precipitations and flows were overestimated in the spring and summer months. Despite this, we accepted them because the differences between the averages were minimised and the flows of the headwaters were better fitted to the observed values (sections 4.2.1 and 4.2.2).

However, we consider that the currently available methods of bias correction may not provide satisfactory fittings to the observed series in this area, so we argue this in more detail in the discussion section, starting with line 518.

The descriptions in 4.2.1 and 4.2.2 correspond to the methods section. The authors should limit to describe the results in these sections.

You are right, these descriptions are now in the Material and methods section and in sections 4.2.1 and 4.2.2 only one sentence was kept to guide the reader through the process, as it may be complicated to understand and lead to confusion.

Moreover, I think the section would benefit from merging figures 7 and 8, although I am not sure whether the first column in figure 8 should be maintained for what I will mention next.

We merged Figures 7 and 8 in the new Figure 5.

Figure 9 shows the mean rates of streamflow change for the three future periods with regard to the reference period. Did the authors check the rate of change between reference and future periods of non-corrected flows from option B? I think this might be revealing. Also, the size difference between graph A and graph B in figure 9 should be revised.

We did not check the average change rates of the non-corrected flows because we thought that the underestimation of the upstream flows was unacceptable, so we decided not to work with them, as we discussed above.

However, the flow change rates are available in the SWICCA portal for each future period, but the results without skill in the re-forecast analysis are found in almost the entire basin. Therefore, we thought that using these change rates would not give us realistic results for the future.

On the other hand, Figure 9 was replaced by the new Figure 6, which shows the same information but represented on the basin maps. We believe this is a very good improvement.

Regarding the final step of the methodology, relative to water management simulation, I think the authors should justify better the added value of using stochastic modeling when they already have a reasonable amount of data (several 30 years series from various climate models) to perform the statistics relative to water storage.

We think that only 9 series (one for each ensemble member) are not enough to apply the risk assessment process, even though we divide them in periods of 30 years. The more equiprobable series we generate with the statistical properties of each ensemble member, the more agreement between them, which means more reliable results. This statement is not shown in the risk indicator (Figure 8) as the ensemble members are quite disperse (dry or wet periods) and they complement each other in the ensemble, resulting in almost the same probabilities of being in any volume interval of the total capacity of the system.

We included lines 304-305 to justify the added value of this step.

I think the results from using the water management simulation model should appear before the ones from the stochastic simulation and, in any case, both results should be comparable (e.g., calculating the drought indicator, or the exceedance probability in September).

Thank you for the suggestion. We placed the results of the water management before those of the risk assessment. Actually, they are already comparable because in both cases we are showing the evolution of the water resources of the system, one in form of risk indicator with probabilities for each future period and in the other with mean volumes of the ensemble and the range covered by it for the entire period.

The exceedance probability at the beginning of the irrigation season might be a relevant result too.

You are right, but we focused on September because it is the end of the irrigation season and the end of the hydrological year. Thus, this result is probably the one that best summarizes the

final state of each campaign. In addition, this is a meaningful data for the stakeholders because it is better understood by them.

However, we included the exceedance probability at the beginning of the irrigation season (March) because it can be a good way of informing the irrigation associations about the possibilities of having shortages and take measures to avoid them.

Thus, we showed both results in the reviewed manuscript.

Anyway, the results of this section show how the relevance of the bias correction is dampened through the modeling chain. Considering limitations and additional uncertainty introduced by bias correction highlighted by Ehret et al. (2013) and the results of studies like Muerth et al. (2013) who argue about the utility of bias correction in model chains, I think the authors lost a good opportunity to contribute to the existing debate on the added value of bias correction in the modeling of climate change impacts on water resources.

You are right, the debate on the added value of bias correction in the modelling of climate change impacts on water resources is very interesting, but that is not one of the purposes of this study. What is important here is that the raw data were very inappropriate in this case, and we applied the most recommended method to correct them. We know that it has its pros and cons, so we decided to test it on the meteorological data and the flows, just in case there was a notable difference between them and to be able to recommend the better option. But in this case the difference between them was not significant.

However, the papers of Ehret et al. (2012) and Muerth et al. (2013) are very interesting, such as the one by Teutschbein and Seibert (2013) which applies the bias correction in different seasons and the one by Switanek et al. (2017) in which the quantile mapping method for climate change applications was improved. We mentioned all of them in the manuscript (introduction and discussion sections) to go a bit further into the subject of bias correction and discuss its application, but as you will understand, we cannot develop this part in detail since the methodology and its adaptation to the case study are more important on this occasion.

I would not like to finalize my review mentioning that the authors make the wrong use of the term tendency throughout the whole text to my understanding. Mostly because the authors do not show whether their results really follow any trend and whether this is significant.

In this case, the term tendency refers to the decrease of flows as we approach 2100, which is shown in the average change rate of the Júcar River Basin (Figure 6). Perhaps the trend is not so evident because the reduction is relatively low (from 1% to 12%), but considering that we are working with the Júcar River system, these decreases can lead to major problems of water deficits and huge economic losses. Therefore, these small decreases are significant, as the basin is already stressed (demands/resources ≈ 0.9) and this presents a big challenge for decision-makers during extreme events such as droughts.

However, we did not use the term "tendency", instead we referred to the "decrease" in order to avoid any misunderstanding.

**Referee #2:**

First, and from the point of view of the structure of the paper, its title is too long and inaccurate. The work contains, in addition to the methodology, a case study and results obtained after applying the developed methodology.

Thank you for the comment.

In the title we wanted to highlight the importance of the methodology, as it integrates the water planning, the risk assessment, and the possibility of making the bias correction in different ways that lead to the characterisation of natural flows. In this case we did not include the case study since it is a methodology that can be applied to other basins, taking into account its features.

However, as you find it convenient, we changed the title with a more concise format and integrating the case study. We considered these options and the first one was the new title of the manuscript:

- Risk assessment in water resources planning under climate change at the Júcar River Basin.
- Characterisation of natural inflows and modelling chain methodologies for risk assessment in water planning under climate change at the Jucar River Basin.

Second, there is confusion between sections 2, Material and methods, and 3, Case study, since subsections 3.1 and 3.2, and perhaps 3.3, would be better classified as Material and methods.

You are right, the case study was now presented in its own section before the Material and methods section, where previous sections 2 and 3 were merged to reduce the length of the manuscript and avoid replication, giving more sense to the structure of the document.

As for figures, figure 1 does not seem necessary and figure 2 is difficult to understand.

Thank you for the suggestion. Figure 1 was introduced as a small clarification that water management and risk assessment are closely related and how each one works, then in Figure 2 we highlighted the key points of each process. However, we remove Figure 1 and its explanation because you are right, it does not seem necessary. Despite this, we included something about this relationship in the introduction and in the case study sections, as it is part of a specialisation related to water resources management in problematic basins such as the Júcar River Basin, which is stressed due to high exploitation rates (demands/resources ≈ 0.9).

Then, we merged Figure 2 and Figure 3 in the new Figure 2, which is now clearer and understandable jointly with its description in the text that clarifies the key points and guides the reader through the adaptation to the basin in the following sections.

The introduction lacks the reference to similar works that have incorporated climate change projections in decision-making processes in other basins, not just those in the Mediterranean environment. And this is important since the results of the work show a great dispersion (see figure 12).

In this part we decided to focus on Mediterranean studies due to the great dispersion of our results. We justified this in the discussion since most of the authors agree that the skill of climate change projections of this area is very low and usually they are not capable of representing the characteristics of historical droughts (lines 501-503, 536-537).

However, in the reformulation of the introduction section, we named similar studies developed in other areas over the world to justify their inherent dispersion, as those developed by Stagl and Hattermann (2016) and Chatterjee et al. (2018) in the Danube River basin and Kansas, respectively. In addition, we also made statements based on studies developed in other European areas, such as Sweden (Teutschbein and Seibert, 2013).

The Material and methods section is quite robust since this work group has implemented numerous modules, already contrasted, in the Aquatool Decision Support System and now used (hydrological model; management model; water allocation model; stochastic model and risk assessment model). This paper provides the integration of climate projections into the model and its impact on future flows in the basin and on the storage of water in the system. In this sense, it uses nine Ensemble members (table 1) that cause a great dispersion of results, as already mentioned, and an inaccuracy in the conclusions. Would it be possible to use only those that have given better results in the Mediterranean region?

Thank you for the comment. As we said in the text (lines 179-181), we decided to use the ensemble provided by the SWICCA portal (Table 1) because they selected the members that were most suitable for the entire Europe. However, their fitting in this area is not good enough to consider only some of them as they are not able to fit perfectly with the observed data. Moreover, they are not capable of reproducing the statistical characteristics or trends in the average year (new Figures 4 and 5) or over the whole period, neither the characteristics of the historical droughts. For this reason we thought that the use of the whole ensemble would provide us with more options and more robustness to the study based on some authors recommendations (lines 59-65), since increasing the number of ensemble members reduces the sampling uncertainty.

Despite all the efforts we made in the first part of the methodology to reduce the uncertainty provided by the RCMs, such as shortening the reference period to be more in line with the current situation of the basin or correcting both the meteorological and the flow data, this was not possible. Therefore, the results are quite dispersed. This reveals the need to improve the skill of climate projections and the use of more sophisticated bias correction techniques, as we said in the discussion section (lines 521-523 and 531-538).

On the one hand, they work with flow data in the basin between 1980-2012 and, on the other hand, the reference period is reduced to the 1980-2000 period. However, as can be seen in Figure 5, there are differences in the average year inflows between the different periods. Can the use of these different periods have an influence on the results obtained?

Thank you for the suggestion. As we said at the end of Section 2, in this basin it is advisable to work with data from the period 1980-2012, since using series with periods prior to 1980 can lead to an overestimation of water resources due to the so-called "effect 80", which is a significant decrease in precipitation and inflows in the basin from the 1980s onwards. Thus, this is the

reason why we decided to shorten the reference period provided by SWICCA (1971-2000) to 1980-2000, in order to try to better represent the current situation of the basin. In this way, we tried to decrease the uncertainty of the magnitude of future changes when we compare future flows with a reference period that represents the basin, such as the results of the average change rates of the whole basin (new Figure 6).

The new Figure 3 shows how the inflows from both periods (1980-2012 and 1980-2000) can be considered equivalent since the difference between their averages is not significant. However, if we had used the period 1971-2000, we would have a different and unrealistic perception of the basin at this time, since water resources are greater than in reality and by correcting future data this would also be transferred to future periods.

Answering your question, yes, the use of the different periods influences the final results, but in this case the difference between using the period 1980-2012 and 1980-2000 is not notable. Indeed, we published a paper (Suárez-Almiñana et al., 2020) that compares the change rates for the whole basin using the three reference periods named before (1980-2012, 1980-2000 and 1971-2000), among others. In that paper we conclude that the average change rates of the basin are very similar when comparing the future flows of each period (2011-2040, 2041-2070, 2071-2098) with the flows of the reference periods 1980-2012 and 1980-2000. However, when those future periods are compared to the reference period 1971-2000, the average change rates are more drastic (up to -23% at the end of the century), which is logical since this period has more resources available, leading to a more extreme and alarmist conclusion than in this case where the average change rates are between -11% and -12% for the whole basin.

Thus, we think that the shortening of the period was a good decision and the differences between the reference periods 1980-2012 and 1980-2000 are not producing very different results.

The results obtained in figures 6, 7 and 8 are only visually compared. In the text it is written, for example, (lines 353-354): "There can be seen how both HBV models results are generally close to the observed flow values". Would it be possible to specify, from a statistical point of view, the term "close"?

In this case, the term "close" refers to the visual distance between their averages, but we included the NSE and PBIAS statistics in the updated version of this manuscript. We calculated these statistics to find out the performance of the RCMs and whether they improved with bias correction, as well as the tailoring of the hydrological model to the basin. To do this we based on the performance ratings recommended by Kalin et al. (2010) (daily time step) and Moriasi et al. (2007) (monthly time step), which were shown in Table 2. In addition, these recommendations were described in sections 3.1.2 and 3.2.2.

The results of figure 9 show a great variability between options A and B, mainly in the two head reservoir, Alarcon and Contreras. In view of the results in Figure 12, could one option be recommended over another?

Thank you for this question. Figure 7 from the updated manuscript shows how the average of the ensemble is lower in option B and the shaded area reaches much lower values than in option A. Therefore, although option B is more dispersed, if we chose it we would be working from the

point of view of security against future intense drought events, which seem to be more frequent and intense in the future. However, we cannot choose one option over the other due to the high dispersion in both cases. All this was exposed in lines 435-440 and 567-570.

Some minor comments would be: - Figure 2: the acronyms of P and T have not been previously defined - Line 133: the acronym RCM is defined later (see line 236) - Lines 223-226: There are several references to geographical names such as the Albufera of Valencia that are not shown on the map in Figure 4.

Thank you for these comments.

We deleted the P and T acronyms because they did not appear so often in the text. The acronym RCM was previously defined and the irrigated crop areas and the wetland belonging to l'Albufera de Valencia were included in the new Figure 1.

**We hope that our responses to the reviewers' comments and the changes we made in the manuscript will be enough to be published in the HESS journal.**

**Definición de estilo:** Texto comentario

**Risk assessment in water resource planning under  climate change at the Júcar River Basin**

[revised manuscript text omitted]

---

## Referee Report (RR1)

Manuscript: Number: HESS-2019-496

**Title: Risk assessment in water resources planning under climate change at the Júcar River**
**Authors: Sara Suárez-Almiñana, Abel Solera, Jaime Madrigal, Joaquín Andreu, Javier Paredes-Arquiola**

Contents overview:
The paper aims to integrate climate change projections into water system management models in order to guide the decision-making taking into account drought risk assessments.
Main results should be in estimating drought risk indicators and management rules in the future for the water resources system (WRS).
A general remark regards the methodological approach of this complex study regarding climate-changes, hydrology, and water system management: paper needs to improve the clarity of exposition in terms of sections arrangement and description of methods. Considering the complexity and the amount of the reported material, I suggest giving at the end of the Introduction an outlook of the following paper content.

Specific Remarks:
1)  The Introduction focus on the need of methodologies to integrate climate projections in the decision process for water management and drought risk assessments in order to evaluate the future impacts on inflows reduction on stored water availability. This aim is stated in (lines 49-51): "That is exactly what we aim to do in this study: proposing a general methodology inspired on the work of Suárez-Almiñana et al. (2017) to integrate climate projections in the decision process throughout a model chain for water management and drought risk assessments, where the future impacts on inflows and water resources are evaluated." Consequently, the paper deals with different modelling approaches (climate projection, storages inflow evaluation, WRS simulation) with a lot of evaluated material, since many years this work group has implemented modules of the Aquatool DSS. As previously stated, a general outlook of paper structure should be given at the end of the introduction. Nevertheless, Introduction is very extended and could be reduced of some information irrelevant (or no strictly requested) for the rest of the manuscript.

2)  Even if the Jucar River Basin is an extensively studied catchment in literature, especially from this research group, I suggest giving some more information regarding the criticality in water system management. At the moment in the paragraph 2 only some data on water stress in the WRS are given. Moreover, as in the Management Plan and in the Drought Management Plan for this basin, the climate projections were not incorporated explicitly (lines 132-135) and in previous studies climate change effects were only assessed by reducing the natural inflows in a certain percentage, it should be interesting to compare previous management rules given in these Plans with the ones obtained using the hereafter proposed procedures

3)  Differences between the two alternatives for characterisation of hydrological models, called option A and option B, are not clearly recognized. The main difference between these alternatives seems to be the application of the bias correction before (option A) or after (option B), nevertheless, future inflows from A and B options are both introduced in the management model to simulate the future water availability (lines 174-75). Consequently, the two series can be considered as different possible runoff scenarios equally probable ? In

any case main statistics of historical and adopted runoff series should be documented in the paper not only graphically and compared with previous values used in Plans.

4) Paragraph 4.3, showing future water storage ensemble results (shaded area) occupies practically the whole field of stored volume in the basin. These results, indicating for authors a huge uncertainty for the future, highlight the opportunity of filtering obtained data in order to provide performances information managing the system. Graduating colours by frequencies could be useful.

5) In 4.4, giving drought risk indicators, the frequency evolution of reservoirs storage in the system can be seen in Fig. 8 for both options A and B and the exceedance probabilities in storage volumes of March and September (Fig. 9). In addition, values of mean allocated resources for demands and consequent deficit values, as well the well known indices of reliability, vulnerability, resiliency derived from the adoption of the Aquatool DSS could be documented.

6) The final phrase in Discussion, pointing out that all the simulations were made taking into account the current conditions of the system should be anticipated in the modelling description paragraph.

---

## Author Response (AR2)

**First of all we would like to thank Referee 1 and Referee 3 for their comments and suggestions. We considered all of them to improve our manuscript. We will respond to the comments point by point, answering them and telling where the changes were integrated in the manuscript.**

**Referee #1:**

The manuscript has certainly improved with regard to the previous version. Most of the suggestions were implemented. Still, I think the authors fail at providing an appropriate description of their methodology. While one can more or less understand what they mean, both the description and figure two are confusing. In addition, the manuscript still requires a thorough language revision. My recommendation is to ask authors to at least improve their description of the methodology and to make figure two clearer.

Thank you for your comments and recommendations.

Despite the efforts we made to make the methodology more understandable in the previous version of the manuscript, it is evident that we failed in the attempt, since it is a complicated process and not very easy to communicate. Therefore, we decided to make figure 2 as simple and understandable as possible. To do this, we divided it into three parts that were described in more detail in the text to clarify the process. We hope that this time the combination between the figure and the text will lead the reader to a better understanding of the proposed methodology. In addition, at the end of the introduction, we clarify the structure of the document to point out what can be found in each section and to which methodological part each section belongs.

Concerning the language revision, we made a linguistic and structural review, but we hope that the English language copy-editing post process of the HESS journal will improve it and solve this problem before its publication.

**Referee #3:**

Contents overview:

The paper aims to integrate climate change projections into water system management models in order to guide the decision-making taking into account drought risk assessments.

Main results should be in estimating drought risk indicators and management rules in the future for the water resources system (WRS).

Thank you for the comment.

In this study, we did not emphasize on management rules for the future since our main objective is to transform climate projections into useful information for decision-making by developing a methodology that can be extended to other basins. In fact, the manuscript highlights the

difficulty of this process and it would not make much sense to estimate management rules for the future, as the results obtained are too uncertain and not conclusive.

A general remark regards the methodological approach of this complex study regarding climate-changes, hydrology, and water system management: paper needs to improve the clarity of exposition in terms of sections arrangement and description of methods. Considering the complexity and the amount of the reported material, I suggest giving at the end of the Introduction an outlook of the following paper content.

Thank you for your suggestion. We completed the introduction with an outlook of the paper content in lines 78-88. Additionally, we modified the description of the methodology (Section 3) and clarified Figure 2 in order to make the process easier and more understandable for the reader.

Specific Remarks:

1) The Introduction focus on the need of methodologies to integrate climate projections in the decision process for water management and drought risk assessments in order to evaluate the future impacts on inflows reduction on stored water availability. This aim is stated in (lines 49-51): "That is exactly what we aim to do in this study: proposing a general methodology inspired on the work of Suárez-Almiñana et al. (2017) to integrate climate projections in the decision process throughout a model chain for water management and drought risk assessments, where the future impacts on inflows and water resources are evaluated." Consequently, the paper deals with different modelling approaches (climate projection, storages inflow evaluation, WRS simulation) with a lot of evaluated material, since many years this work group has implemented modules of the Aquatool DSS. As previously stated, a general outlook of paper structure should be given at the end of the introduction. Nevertheless, Introduction is very extended and could be reduced of some information irrelevant (or no strictly requested) for the rest of the manuscript.

As mentioned above, we completed the introduction with an outlook of the paper content in lines 78-88. Moreover, we reduced the introduction by removing some references to climate services and related studies since, as you said, it might be irrelevant for the rest of the manuscript.

2) Even if the Jucar River Basin is an extensively studied catchment in literature, especially from this research group, I suggest giving some more information regarding the criticality in water system management. At the moment in the paragraph 2 only some data on water stress in the WRS are given. Moreover, as in the Management Plan and in the Drought Management Plan for this basin, the climate projections were not incorporated explicitly (lines 132-135) and in previous studies climate change effects were only assessed by reducing the natural inflows in a certain percentage, it should be interesting to compare previous management rules given in these Plans with the ones obtained using the hereafter proposed procedures.

Thank you for the recommendation.

We modified the case study section to clarify the criticality of water management in this system by including the total volume of the demands, the reservoirs operation (lines 121-125), as well as the problems within the drought events and some measures to face them (lines 130-133).

Concerning the management rules, we did not mention them in this study, as stated before, but it is evident that some changes should be expected from the results of this type of studies. Therefore, we made a statement related to the revision of the operational rules in the discussion section (lines 605-608).

3) Differences between the two alternatives for characterisation of hydrological models, called option A and option B, are not clearly recognized. The main difference between these alternatives seems to be the application of the bias correction before (option A) or after (option B), nevertheless, future inflows from A and B options are both introduced in the management model to simulate the future water availability (lines 174-75). Consequently, the two series can be considered as different possible runoff scenarios equally probable? In any case main statistics of historical and adopted runoff series should be documented in the paper not only graphically and compared with previous values used in Plans.

Thank you for the comment.

These options are simply two different ways of working with the same data. Thus, depending on the decision made, different results can be obtained, that is why both followed the same process to the storage and risk assessments. This is only to know which alternative can provide more reliable results at the end of the process. Therefore, within these options we are trying to deduce which is more reliable or effective for the purposes of the study. We clarified this in the methodology section (lines 179-181).

However, your question is a point to consider, if there were no favourable conclusion, one option is to keep both for the same analysis, and thus work with more generations using them as equiprobable scenarios to get only one result.

Concerning the main statistics of the generated and observed series and their comparison, they are shown indirectly in Table 2 and lines 357-362, since both data were used to extract those statistics. In fact, some of these data can be found documented numerically in the document of Suárez-Almiñana et al. (2020). In this case, we preferred not to add more details about it due to the length of the manuscript.

4) Paragraph 4.3, showing future water storage ensemble results (shaded area) occupies practically the whole field of stored volume in the basin. These results, indicating for authors a huge uncertainty for the future, highlight the opportunity of filtering obtained data in order to provide performances information managing the system. Graduating colours by frequencies could be useful.

Thank you for the recommendation.

We modified Figure 7 by including a lighter area in the ensemble, which is related to the frequency zone of the EMs. To do this we remove the 2 minimum and maximum values of the ensemble and the result was commented in the text, lines 460-463.

5) In 4.4, giving drought risk indicators, the frequency evolution of reservoirs storage in the system can be seen in Fig. 8 for both options A and B and the exceedance probabilities in storage volumes of March and September (Fig. 9). In addition, values of mean allocated resources for demands and consequent deficit values, as well the well known indices of reliability, vulnerability, resiliency derived from the adoption of the Aquatool DSS could be documented.

We considered the inclusion of some of the results you named, as they are usually very valuable, but this time and taking into account the uncertainty of the results, we decided to show the most relevant results from our point of view, as the ranges covered by the ensemble are very wide. Thus, the conclusion reached in this manuscript would be the same. In addition, we did not want to increase the length of the manuscript, it may be too tedious for the reader.

6) The final phrase in Discussion, pointing out that all the simulations were made taking into account the current conditions of the system should be anticipated in the modelling description paragraph.

Thank you for the suggestion. We anticipated this statement in the modelling description section, lines 196-197.

**We hope that our responses to the reviewers' comments and the changes we made in the manuscript will be enough to be published in the HESS journal.**

[revised manuscript text omitted]

**Characterization of natural inflows**

Climate change projections
(precipitation & temperature)

**A**

*Bias correction*
(precipitation & temperature)

**Hydrological model**

**Hydrological model**

*Bias correction*
(inflows)

**B**

**Future inflows**
(A option)

**Future inflows**
(B option)

**Deterministic approach**

**Future inflows**
(A & B options)

**Management model**
(water allocation model)

**Future water storage**
in the system

**Probabilistic approach**

**Statistical properties from future inflows**
(A & B options)

**Stochastic model**

**Risk assessment model**

**Drought Risk Indicator**
(probability of reservoir storage)

[revised manuscript text omitted]